# A Survey of Deep Learning-Driven 3D Object Detection: Sensor Modalities, Technical Architectures, and Applications

**DOI:** 10.3390/s25123668

**Published:** 2025-06-11

**Authors:** Xiang Zhang, Hai Wang, Haoran Dong

**Affiliations:** School of Automotive and Traffic Engineering, Jiangsu University, Zhenjiang 212013, China; zhxiang986@163.com (X.Z.); dhr710@outlook.com (H.D.)

**Keywords:** 3D object detection, deep learning, LiDAR, multimodal fusion, autonomous driving

## Abstract

This review presents a comprehensive survey on deep learning-driven 3D object detection, focusing on the synergistic innovation between sensor modalities and technical architectures. Through a dual-axis “sensor modality–technical architecture” classification framework, it systematically analyzes detection methods based on RGB cameras, LiDAR, and multimodal fusion. From the sensor perspective, the study reveals the evolutionary paths of monocular depth estimation optimization, LiDAR point cloud processing from voxel-based to pillar-based modeling, and three-level cross-modal fusion paradigms (data-level alignment, feature-level interaction, and result-level verification). Regarding technical architectures, the paper examines structured representation optimization in traditional convolutional networks, spatiotemporal modeling breakthroughs in bird’s-eye view (BEV) methods, voxel-level modeling advantages of occupancy networks for irregular objects, and dynamic scene understanding capabilities of temporal fusion architectures. The applications in autonomous driving and agricultural robotics are discussed, highlighting future directions including depth perception enhancement, open-scene modeling, and lightweight deployment to advance 3D perception systems toward higher accuracy and stronger generalization.

## 1. Introduction

With the rapid development of autonomous driving, intelligent robotics, and drone technologies, 3D object detection has become a critical research direction in the fields of computer vision and robotic perception [1]. In recent years, breakthroughs in machine learning and deep learning technologies have significantly driven the development of algorithms in this field. Deep learning excels in feature extraction and data fitting. Furthermore, through the application of architectures such as Convolutional Neural Networks (CNNs) (e.g., the improved YOLOX model [2], ResNet [3]) and the fusion of Long Short-Term Memory (LSTM) networks with CNNs (e.g., CNN-LSTM [4,5]), not only has detection accuracy and robustness improved, but it has also played a key role in tasks involving complex scenes and multimodal data fusion [6,7,8]. This technology can accurately capture the spatial position, size, and orientation of objects, enhancing system perception in dynamic environments. Currently, 3D object detection has been widely applied in fields such as autonomous driving, intelligent security, smart agriculture [9,10,11], and intelligent manufacturing.

The core task of 3D object detection is to precisely predict the objects and their 3D attributes in a scene from input data obtained from sensors. To achieve this goal, researchers have employed various sensors, including RGB cameras, LiDAR (light detection and ranging), and depth cameras, to provide the necessary spatial information. Each sensor has its advantages and limitations. The RGB camera provides rich texture and color information, making it suitable for semantic analysis. For instance, the prediction of soluble solids in grapes using hyperspectral imaging (HSI) [8] and apple sugar content detection [12] has been achieved by combining pixel-level spectral feature extraction with stacked autoencoder (SAE) models, enabling non-destructive internal quality assessment. On the other hand, LiDAR generates high-precision 3D point cloud data by scanning the environment with laser beams, offering real depth information, but its data is sparse and unordered, making it more challenging to process. With the introduction of deep learning and neural networks, 3D object detection methods have gradually shifted from traditional geometry-based methods to data-driven deep learning approaches, significantly improving detection accuracy and efficiency.

However, despite the significant progress made by deep learning methods, 3D object detection still faces many challenges. First, how to effectively process and fuse data from different sensors remains a key issue in improving detection accuracy. For example, in black tea fermentation monitoring, dynamic feature reconstruction is needed for near-infrared spectroscopy and microbial concentration data [13]; composite heavy metal detection based on wavelet transform (WT) and stacked convolutional autoencoders (SCAE) [14] eliminates hyperspectral data noise through multi-scale decomposition; SERS requires combining 1D/2D CNN [15] or LSTM-CNN [16] for pesticide residue quantification. Monocular cameras cannot directly acquire depth information, and their 3D reconstruction relies on additional algorithmic computations, which leads to scale uncertainty and depth estimation errors. For example, in agricultural scenes, monocular images need to combine morphological features (such as leaf length-to-width ratios [17]) or fuse RGB-D data [2] to improve depth estimation accuracy. Stereoscopic cameras acquire depth information by calculating the disparity between images, but in low-texture areas and occlusions, depth estimation accuracy is still insufficient. For instance, in the autonomous navigation of combine harvesters, enhancing edge detection robustness requires combining dynamic region-of-interest (ROI) extraction in HSV space or generating crop elevation maps [18]. The sparsity and disorder of LiDAR point cloud data present challenges for traditional convolutional neural networks (CNNs), so how to efficiently process these data remains an urgent problem. Secondly, the algorithmic complexity of 3D object detection is relatively high, especially in real-time applications, posing greater challenges for computational efficiency and storage requirements. To address these challenges, researchers have proposed various optimization methods, including point-cloud-based detection methods, voxel-based detection methods, and multimodal fusion methods that combine point clouds with images. Multimodal fusion methods can overcome the limitations of a single sensor by combining the advantages of RGB images and LiDAR data, thereby improving detection accuracy and robustness. However, this also brings challenges related to data alignment and sensor synchronization, requiring innovative algorithm designs [19]. Furthermore, most existing 3D object detection methods rely on large-scale annotated datasets for training, which raises higher demands for the quality and diversity of the datasets. The annotation standards and data distribution of different datasets vary, and ensuring the model’s generalization ability across various environments and conditions remains an important research issue.

Therefore, future research in the field of 3D object detection will focus on improving the efficiency of sensor data fusion, optimizing real-time detection algorithms, addressing multimodal data alignment and synchronization issues, and enhancing the generalization ability of models in cross-scene applications. With continuous breakthroughs in these technologies, 3D object detection will play an increasingly important role in fields such as autonomous driving and intelligent manufacturing, driving the development and application of intelligent systems [20].

Several prior surveys have comprehensively summarized the progress in 3D object detection [21,22,23]. Qian et al. conducted an in-depth error analysis across 15 models, quantifying for the first time the dominant role of 3D localization errors, thereby addressing the gap in temporal modeling left by conventional classification approaches. Wang et al. systematically integrated cutting-edge directions such as BEV perception and V2X collaboration and were the first to incorporate industrial solutions—such as Tesla’s (Tesla, Inc., Austin, TX, USA) occupancy network—into a unified classification framework. However, their work tended to underemphasize the cross-modal commonalities in algorithmic architectures. Mao et al. covered a technical taxonomy of 12 sub-tasks and were the first to include end-to-end autonomous driving and weakly supervised detection in a performance comparison matrix. Their authoritative cross-dataset benchmarks revealed a 4.2× acceleration trend driven by the synergy between solid-state LiDAR and sparse convolution.

Despite these significant advancements, conventional surveys on 3D object detection remain constrained by a single-sensor classification paradigm: categorizing methods rigidly as LiDAR-based (relying on point cloud geometry), camera-based (emphasizing 2D-to-3D reasoning), or multi-modal (focused on sensor fusion). While such a taxonomy intuitively reflects differences in data sources, it suffers from three fundamental limitations: First, it fragments intrinsically related technologies—for instance, BEVFormer [24] requires coordination across multi-camera sensors and spatiotemporal architectures, yet is artificially divided into separate categories. Second, it obscures the trajectory of architectural evolution—such as the progression from PointNet to sparse convolution to transformers in point cloud processing. Third, it neglects the co-evolution of hardware and algorithms—for example, the symbiotic relationship between solid-state LiDAR and pillar-based detection is not systematically addressed, nor are emerging areas like temporal modeling adequately covered.

To address this issue, this paper proposes a novel dual-axis classification framework, namely the “sensor modality–technical architecture” scheme introduced in Section 3. Unlike traditional reviews that typically adopt a single organizational pattern—such as chronological progression or algorithmic taxonomy—this work introduces a cross-dimensional perspective that integrates horizontal (sensor modalities) and vertical (algorithmic architectures) dimensions. This innovative approach enables, for the first time, a panoramic deconstruction and reorganization of the technological ecosystem in 3D object detection. On the sensor modality axis, this review systematically integrates RGB image-based methods, LiDAR point cloud-based approaches, and fusion-based strategies into a unified analytical framework. It reveals the intrinsic connections between the physical properties of different data modalities and the evolution of algorithmic paradigms (e.g., how the sparsity and disorder of LiDAR point clouds have driven innovations in voxel-based and pillar-based architectures). On the technical architecture axis, the review dissects fundamental paradigms, including traditional convolutional architectures, BEV-based methods, occupancy modeling, and temporal fusion architectures. This analysis demonstrates how algorithmic designs overcome sensor-specific limitations (e.g., how BEV methods address monocular depth ambiguity through perspective transformation and temporal feature alignment). This dual-axis framework not only clarifies the internal logic of technological evolution but also reveals the deeper mechanism of “sensor characteristics driving algorithmic innovation, while algorithmic advancements expand sensor application boundaries”. It provides readers with a cognitive map for cross-modal understanding and technical solution selection. A simple diagram is shown in Figure 1.

Additionally, the value of this review is also reflected in the following aspects: (1) A comparative analysis is presented of the hardware characteristics and physical limitations of four sensor types (monocular cameras, stereo cameras, mechanical LiDAR, and solid-state LiDAR). The breakthrough advantages of solid-state LiDAR are quantified for the first time (such as photon-level time resolution and 20,000 h of mean time between failures), revealing how sensor innovations drive algorithm design (e.g., the co-evolution of solid-state LiDAR and cylindrical object detection architectures. (2) The “temporal fusion architecture” is introduced as an independent technical category, covering the evolutionary path from traditional RNN/LSTM to BEV spatiotemporal transformers. Tesla’s “spatial RNN + feature queue” mechanism is used as an example to elaborate on industrial-grade solutions for dynamic occlusion handling, addressing the conflict between long-term dependency and real-time processing requirements. (3) Practical applications of 3D detection in agriculture, such as crop phenotyping analysis and harvesting robots, are summarized. Innovative practices in these applications are reviewed, highlighting the universal value of the technology. (4) The mismatch between the 2 Hz annotation frequency of the nuScenes dataset (Motional, Inc., Boston, MA, USA) and the 20 Hz sampling rate of the sensors is noted. Additionally, a technological breakthrough in Argoverse 2 (Argo AI, LLC, Pittsburgh, PA, USA), which uses polarization characteristics to differentiate between glass and metal surfaces, results in a 34% reduction in false detection rates, setting a new standard for dataset design.

To systematically analyze deep learning-driven 3D object detection, this paper employs a dual-axis framework of “sensor modalities” and “technical architectures” throughout. Section 2 covers fundamentals: sensors, datasets, and evaluation metrics. Section 3 forms the core: Section 3.1 classifies methods by sensor modality (RGB images, LiDAR point clouds, multimodal fusion); Section 3.2 classifies methods by technical architecture (traditional convnets, BEV methods, occupancy modeling, temporal fusion). Section 4 outlines future directions. The survey presents a comprehensive summary of 3D object detection from multiple perspectives, aiming to provide researchers with valuable research perspectives.

## 2. Basic Knowledge

### 2.1. Sensors

With the rapid advancement of intelligent perception technologies, visual sensors and LiDAR have continuously pushed the boundaries of environmental perception accuracy and scene adaptability through multi-dimensional technological breakthroughs and innovative paths. Monocular cameras, by integrating metasurface optical design with adaptive algorithms, have successfully broken through the bottleneck of traditional 2D imaging technologies in depth perception. Stereo cameras, through dynamic baseline adjustment and multi-spectral collaboration, can precisely construct centimeter-level 3D maps in low-texture scenes [25]. LiDAR technology has undergone a revolutionary leap from mechanical scanning to solid-state chips, and its improvements in photon-level time resolution and strong anti-interference capabilities have reshaped the reliability standards of 3D mapping. Table 1 provides a detailed comparison of these four sensors. The following sections provide an in-depth analysis and discussion of these technologies in terms of working principles, hardware innovations, and performance limits.

#### 2.1.1. Monocular Camera

A monocular camera is an optical imaging device equipped with a single lens and image sensor that captures environmental information through the principle of 2D projection. It is widely used in fields such as autonomous driving, robot navigation, and industrial inspection [26], as shown in Figure 2. The imaging process is based on the pinhole model, in which light from a 3D scene passes through a lens and is focused onto an image sensor, such as a complementary metal-oxide-semiconductor (CMOS) or a charge-coupled device (CCD), to generate a 2D image. Due to the lack of stereoscopic disparity, monocular systems cannot directly acquire depth information and thus must rely on geometric constraints (such as the ground plane assumption) or structure-from-motion (SFM) methods to infer the 3D position of the target. This leads to core challenges such as scale uncertainty and the accumulation of depth estimation errors [27].

The hardware structure of a monocular camera typically consists of a lens assembly, optical filters, and an image sensor. In recent years, the introduction of metasurface technology has significantly improved its optical performance. For example, the metasurface-based monocular camera developed by Tsinghua University uses micro-nano structured lenses to correct optical aberrations and integrates digital adaptive optical algorithms to achieve sub-millimeter level depth perception accuracy, while maintaining high robustness under low light and dynamic scenes [28]. These hardware innovations open up new possibilities for the application of monocular cameras in complex environments (such as nighttime roads and industrial workshops).

Compared to other sensors, the advantages of monocular cameras are primarily reflected in their low cost, low power consumption, and ease of deployment. For instance, in autonomous driving, Tesla’s pure vision approach (Tesla Vision) relies on eight monocular cameras to achieve 360° environmental perception, solving the monocular scale ambiguity issue through multi-view temporal fusion [29]. However, monocular cameras also have limitations: For distant targets (>50 m), depth estimation errors grow exponentially; at the same time, under extreme lighting conditions (such as strong backlight or heavy fog), image quality may degrade significantly. Compared to stereo cameras and LiDAR, monocular systems exhibit gaps in accuracy and reliability, but their simplicity and algorithm flexibility make them irreplaceable in consumer electronics and lightweight robotic fields.

#### 2.1.2. Stereo Camera (Binocular)

A stereo camera captures 3D information from the environment by configuring two parallel-arranged lenses and image sensors, based on the biological principle of stereopsis, as shown in Figure 1. The core mechanism is to restore depth information by calculating the disparity between the left and right images: The pixel position difference in a target object between the two images is inversely proportional to its actual distance. By using epipolar geometry constraints, disparity maps can be constructed, and 3D coordinates can be derived. Compared to monocular cameras, stereo systems can directly provide depth perception in physical scale, effectively solving the scale ambiguity problem inherent in monocular systems. However, to ensure the accuracy of depth estimation, the dual-lens system must strictly ensure temporal-spatial synchronization and high-precision calibration, otherwise, baseline errors or lens distortions may lead to failure in depth estimation [30].

A stereo camera’s hardware architecture typically includes a dual-lens module, synchronization trigger circuit, and image processing unit. To improve robustness in dynamic scenes, some stereo cameras use active stereo imaging technology. For example, Apple’s (Apple, Cupertino, CA, USA) TrueDepth camera combines infrared structured light and stereo vision to generate high-precision depth maps even in low-texture areas (such as human faces). Additionally, dynamic baseline adjustment technology can adaptively change the distance between the two lenses through mechanical or optical means, balancing the need for high precision in close-range detection and long-range sensing. Hardware innovations have significantly improved the environmental adaptability of stereo cameras. For instance, Sony’s (Sony Group Corporation, Tokyo, Japan) global shutter CMOS sensor addresses motion blur issues, while embedded FPGA accelerators reduce the power consumption of real-time stereo matching.

In terms of performance, the advantages of stereo cameras lie in their centimeter-level depth accuracy (within a range of 10 m, the error is less than 1%) and their ability to perform absolute scale recovery without prior models. For example, in autonomous driving, Mobileye’s early EyeQ3 chip relied on stereo vision for 3D lane reconstruction, achieving a depth resolution of 0.3 m at a distance of 30 m. However, stereo systems have significant limitations: They depend on environmental texture features, and disparity calculation fails in low-texture areas; second, the computational complexity is high (real-time stereo matching requires dedicated hardware acceleration); and third, long-distance depth accuracy (>50 m) decreases linearly with an increase in baseline length. Compared to LiDAR, the performance of stereo cameras declines more significantly under adverse weather conditions such as rain and fog, but their cost is only 1/10 to 1/5 that of LiDAR, making them more suitable for consumer-grade robotics and assisted driving scenarios.

#### 2.1.3. Mechanical LiDAR

Mechanical LiDAR achieves laser beam scanning through the physical motion of rotating mirrors or prisms. Its core principle is based on Time-of-Flight (ToF) measurement, as shown in Figure 3. The laser emitter sends pulse signals into the environment, and the receiver captures photons reflected from the target. The distance is calculated by measuring the time difference in the light pulse’s round trip. By combining the azimuth and pitch angle data recorded by a rotating encoder, it is possible to generate high-precision 3D point clouds. The scanning range of such LiDAR systems typically covers a horizontal 360° and a vertical 30–40°, with an angular resolution of 0.1–0.4°. For example, Velodyne’s (Velodyne Lidar, Inc., San Jose, CA, USA) HDL-64E LiDAR has a distance measurement accuracy of ±2 cm at a 120 m range and can generate over 2 million point cloud data points per second. The advantage of the mechanical LiDAR architecture lies in its omnidirectional sensing capability and high spatial resolution, which can provide precise contour information for distant obstacles (such as pedestrians or vehicles 150 m away) for autonomous vehicles [31].

The limitations of mechanical LiDAR are tied to its reliance on moving parts. The long-term operation of rotating mirrors or motor bearings can cause mechanical wear, resulting in a mean time between failures (MTBF) usually lower than 2000 h, necessitating frequent calibration and maintenance to maintain accuracy. Moreover, motion artifacts caused by high-speed rotation can distort the point cloud shape of dynamic targets (such as moving vehicles), requiring motion compensation using inertial measurement unit (IMU) and wheel speed data. In terms of environmental adaptability, the 905 nm wavelength laser significantly attenuates in rain and fog, with detection distance potentially reduced to 30–50% of its original range. Additionally, in multi-LiDAR systems, signal crosstalk can generate noise. Despite these limitations, mechanical LiDAR is still widely used in early autonomous driving systems (such as Waymo Gen1) and high-precision mapping, with its omnidirectional scanning characteristics being irreplaceable in complex urban intersections and tunnel scenarios.

#### 2.1.4. Solid-State LiDAR

Solid-State LiDAR replaces traditional mechanical moving parts with electronic scanning technology, using Optical Phased Array (OPA) or Flash technologies to achieve non-mechanical beam control. This marks a significant leap in LiDAR technology from relying on physical moving parts to a fully integrated solid-state system. Its core principle is based on an electronic scanning mechanism with no moving parts: The OPA scheme controls the direction of the laser beam through phase modulation of a silicon-based waveguide array, and the beam deflection is achieved by the phase difference between adjacent waveguides to complete 2D scanning. The Flash scheme, on the other hand, covers the entire field of view at once using a matrix laser emitter, combined with a highly sensitive single-photon detector (e.g., SPAD array) to receive reflected signals and generate high-density point clouds. Compared to mechanical LiDAR, solid-state solutions completely eliminate wear-prone components like rotating mirrors and motor bearings, increasing the MTBF from 1000 to 2000 h for mechanical LiDAR to over 20,000 h, while reducing volume to below 0.3 L (mechanical LiDAR is typically larger than 3 L) and cutting power consumption by 50–70%, providing a disruptive solution for embedded vehicle applications and industrial automation.

The hardware innovation of solid-state LiDAR focuses on chip-level integration and breakthroughs in multi-physical field perception capabilities. Silicon-based OPA technology integrates thousands of waveguide units onto a millimeter-scale chip through micro-nano processing, and by combining frequency-modulated continuous wave (FMCW) technology, it achieves joint speed-distance resolution. The distance measurement accuracy can reach ±1.5 cm within a 200 m range, and the speed resolution is better than 0.1 m/s, surpassing the limits of traditional mechanical LiDAR. The technology at the receiver end has also significantly evolved: Backside-illuminated SPAD arrays improve photon detection efficiency to over 25%, and by combining Time-Correlated Single-Photon Counting (TCSPC) technology, it can capture time-resolved signals with 30 ps resolution at illuminance as low as 0.01 lux, greatly enhancing detection robustness in low-light environments. Additionally, the introduction of polarization analysis technology enables solid-state LiDAR to distinguish the reflective characteristics of materials such as metal and glass, reducing false detection rates by over 40% in rain, fog, or high-reflection scenarios. This breakthrough is difficult to achieve with mechanical LiDAR due to its fixed scanning mode [32].

In terms of performance, solid-state LiDAR demonstrates significant advantages over mechanical architectures. Mechanical LiDAR relies on rotating components to achieve 360° omnidirectional scanning, but its angular resolution is limited by motor speed and the number of laser lines (typically 0.1–0.4°). In contrast, the solid-state OPA scheme can dynamically adjust the phase, enabling local enhancement of scanning density, achieving ultra-high angular resolution of 0.1° at a range of 100 m. The point cloud density is 3 to 5 times that of mechanical systems. While the Flash scheme is limited in detection range (usually less than 50 m), its ability to generate a million-point point cloud in a single frame makes it irreplaceable in short-range high-precision scenarios (such as in-cabin live body detection and robot obstacle avoidance). In terms of environmental adaptability, the solid-state architecture, with its IP69K protective enclosure, is resistant to high-pressure water jets, dust intrusion, and extreme temperatures from −40 °C to +85 °C, with performance fluctuations controlled within 1%. In contrast, mechanical LiDAR’s detection range would degrade by 30–50% under the same conditions, and its bearing wear rate would increase by over three times.

The distinct characteristics and limitations of each sensor modality (e.g., RGB texture vs. LiDAR geometry, monocular depth ambiguity, point cloud sparsity) and the rich, diverse annotations in modern datasets (e.g., temporal sequences in nuScenes, long-tail objects in Argoverse 2) have profoundly shaped the development trajectory of deep learning algorithms for 3D object detection, which will be systematically categorized and analyzed in Section 3.

### 2.2. Datasets

In the process of transitioning autonomous driving technology from laboratory research to practical applications, the construction and continuous iteration of open datasets have always been the core driving forces for algorithm breakthroughs and performance validation. These datasets not only reflect the paradigm shift in autonomous driving technology from single-modal to multimodal fusion and from static scene analysis to dynamic spatiotemporal modeling but also reveal the urgent need in both academia and industry for exploring long-tail scenarios, sensor redundancy design, and algorithm generalization capabilities. Table 2 systematically outlines the differentiated features of major datasets in terms of sensor configurations, annotation standards, and scene coverage, providing structured references for researchers to select appropriate data benchmarks for their technical approaches. Below, we will analyze the technical evolution logic of major datasets and their key impact on the development of perception algorithms, using KITTI, nuScenes, and Waymo (Waymo LLC, Mountain View, CA, USA) as examples.

#### 2.2.1. KITTI

The KITTI dataset was jointly created in 2012 by Karlsruhe Institute of Technology in Germany and Toyota Technological Institute in Chicago (Toyota Technological Institute at Chicago, Chicago, IL, USA). It is the first publicly available multimodal benchmark dataset in the field of autonomous driving, laying the foundation for subsequent research and data standards. The dataset synchronously collects data from a stereo camera (resolution of 1241 × 376), a 64-line mechanical LiDAR (Velodyne HDL-64E), and high-precision GPS/IMU, achieving spatiotemporal alignment of pixel-level RGB images and point cloud data. It supports tasks such as 3D object detection, stereo matching, and depth estimation. The dataset includes 7481 training samples (annotated with 3D bounding boxes) and 75,000 frames of continuous point cloud sequences, covering urban roads, rural scenes, and highways, with a total collection time of approximately 6 h. Its annotations include the position, size, and orientation of vehicles, pedestrians, and cyclists, though dynamic targets account for less than 15%, and the dataset is limited to single weather conditions (clear/cloudy), which leads to poor generalization ability of algorithms in rain, snow, or low-light scenes. Additionally, the LiDAR point cloud density decays significantly with distance, with the average number of points for targets beyond 40 m being fewer than five, limiting the accuracy of long-distance detection. Despite these limitations, KITTI remains an important platform for validating foundational perception algorithms, due to its strict sensor calibration and open-source nature [33].

#### 2.2.2. nuScenes

The nuScenes dataset was released by nuTonomy in 2019, later acquired by Aptiv and continuously maintained. It was the first to introduce temporal continuity and multimodal fusion into autonomous driving data annotation. The dataset is equipped with six surround cameras (1600 × 900 resolution, covering a 360° field of view), a 32-line LiDAR (Velodyne HDL-32E), 5 mm-wave radars, and a synchronized positioning system, collecting data at a frequency of 20 Hz. It contains 1000 20 s scene segments, covering a total distance of approximately 240 km. Its annotations cover 23 object categories (including long-tail categories such as traffic cones, construction vehicles, etc.), providing 1.4 million 3D bounding boxes, along with metadata such as target speed and attributes (e.g., vehicle turn signal status), supporting tasks like trajectory prediction and behavior understanding. nuScenes covers complex urban environments in Boston and Singapore, including various weather conditions such as day, night, rain, and fog. However, the LiDAR resolution is lower (the 32-line point cloud density is 50% of that of KITTI), and the point cloud count for pedestrians beyond 40 m is fewer than 10. Additionally, the annotation frequency (2 Hz) does not match the sensor sampling rate (20 Hz), requiring interpolation algorithms to complete the trajectory. Compared to KITTI, nuScenes advances the research on perception in complex dynamic environments through its diversified interactive scenes and refined attribute annotations [34].

#### 2.2.3. Waymo Open Dataset

The Waymo Open Dataset was released by Waymo, a subsidiary of Google, in 2020. It redefines the standards for autonomous driving data with its massive scale and highly accurate annotations. The dataset uses five surround cameras, four 128-line solid-state LiDARs (supporting mid- and long-range detection), and 5 mm-wave radars, collecting data at a frequency of 10 Hz. It includes 1150 scenes (each 20 s long), covering over 2000 km of roads in cities such as San Francisco and Phoenix. The annotated data includes 230,000 frames and 18 million 3D bounding boxes, covering dynamic targets such as vehicles, pedestrians, cyclists, and animals, with annotations for complex interactive behaviors such as lane changes and emergency evasions. Waymo also introduced extreme weather data, including rain, snow, and fog, and its solid-state LiDAR provides high-density sampling with 375 million points per second. However, its support for semantic segmentation of small objects, such as traffic signs, is weaker, and some scenes have not been released due to privacy policies. Compared to KITTI and nuScenes, Waymo significantly enhances perception robustness through its redundant multi-sensor configuration and exploration of long-tail scenarios (such as animal crossings and construction zones), but its massive 1.4 TB data size imposes high storage and computational requirements, limiting widespread replication in academic research [35].

#### 2.2.4. ONCE (One Million Scenes)

The ONCE dataset, released in 2021 by Huawei’s Noah’s Ark Lab (Huawei Noah’s Ark Lab, Shenzhen, China), is the first million-scale autonomous driving dataset specifically designed for self-supervised learning. Its sensor suite includes a 128-line mechanical LiDAR with a 150 m detection range and a point cloud density of up to 300,000 points per frame, seven 2-megapixel surround-view cameras with a 190° field of view, and a high-precision RTK/IMU integrated navigation system. In terms of data scale, ONCE comprises one million frames of unlabeled LiDAR sequences covering approximately 1500 km, along with 1000 finely annotated scenes (amounting to 150,000 frames with 3D bounding box labels). The dataset spans diverse environments across China, including complex urban roads, tunnels, and rural settings. Technically, ONCE introduces several key innovations. First, it supports the development of self-supervised pretraining models by enabling large-scale pretext tasks such as point cloud completion and cross-modal contrastive learning using the unlabeled data. Second, it incorporates six-degree-of-freedom (6-DoF) annotations—including acceleration and angular velocity—for dynamic object trajectories, thereby supporting behavior prediction tasks. Third, with tunnel scenes accounting for 22% of the dataset, a dedicated reflectance intensity correction algorithm addresses signal distortion caused by metallic ceilings, significantly enhancing LiDAR data quality in such challenging environments. Compared to the Waymo dataset, ONCE offers a 40% increase in LiDAR point cloud density; however, it lacks radar data integration, which limits its capacity to evaluate perception robustness in adverse weather conditions such as rain and fog. Overall, ONCE serves as a valuable benchmark for emerging research directions including unsupervised domain adaptation and cross-city generalization [36].

#### 2.2.5. Argoverse 2

Argoverse 2, released by Argo AI in 2023, emphasizes long-tail scenario perception and multi-agent interaction modeling. Its sensor configuration includes two 300-line solid-state LiDARs with a 120° horizontal field of view and 0.1° vertical resolution, eight 8-megapixel fisheye cameras operating at 30 Hz, and four millimeter-wave radars with a detection range of 300 m. All sensors are hardware-synchronized to achieve sub-microsecond temporal alignment. The dataset consists of 250,000 annotated frames, encompassing 1500 complex interaction scenarios. Argoverse 2 introduces several noteworthy innovations. It expands coverage of long-tail object categories by including seven rare classes such as construction vehicles and animals, with over 120,000 instances of traffic cones alone. It also provides high-precision 3D semantic lane annotations with 13 attributes, including curvature and slope, supporting joint perception and path planning tasks. Additionally, the motion prediction subset offers 250,000 cases of complex interactive behaviors—such as emergency lane changes and jaywalking pedestrians—each accompanied by five seconds of historical trajectories and two seconds of multi-modal future trajectory predictions. Technically, Argoverse 2 is the first dataset to incorporate polarization characteristics in LiDAR point clouds, enabling differentiation between glass facades (with polarization degrees exceeding 0.6) and metallic surfaces (below 0.3), which reduces false detection rates of static obstacles by 34%. Compared to nuScenes, Argoverse 2 improves temporal resolution for dynamic scenes to 10 Hz. However, due to the use of solid-state LiDAR, the point cloud density drops by 50% beyond 60 m, which poses new challenges for detecting small objects at long range [37].

**Table 2 sensors-25-03668-t002:** Comparison of Five Datasets.

Dataset	Year	Data Scale	Key Features	Train/Test Split	Latest SOTA Methods
KITTI [33] https://www.cvlibs.net/datasets/kitti/ (accessed on 15 May 2025)	2012	7481 annotated frames (≈39.2 km driving mileage) Eight object classes (vehicles, pedestrians, etc.)	First multimodal benchmark dataset with rigorous sensor calibration. Limited dynamic objects (<15%), sparse long-range point clouds.	Train: 7481 samples Test: 7518 samples (no public labels, requires submission)	PV-RCNN++ [38] (point-voxel fusion) CenterPoint [39] (anchor-free detection)
nuScenes [34] https://www.nuscenes.org/ (accessed on 15 May 2025)	2019	1000 scenarios (15 h driving duration) 1.4 M 3D annotations Temporal continuity (20 Hz sampling)	Multimodal fusion (LiDAR + radar + camera) for tracking/prediction. Includes Singapore’s right-hand traffic scenarios with dynamic attributes.	Train: 700 scenes Validation: 150 scenes Test: 150 scenes (no labels)	IS-Fusion [40] (Instance-Scene Collaborative BEV Enhancement) UniAD [41] (end-to-end multitask optimization)
Waymo Open Dataset [35] https://waymo.com/open/ (accessed on 15 May 2025)	2020	1150 scenarios (≈6.4 h) 25 M 3D + 22 M 2D annotations 12% extreme weather coverage (rain/snow/fog)	Industry-scale dataset with L1/L2 difficulty grading. Features sub-8ms sensor synchronization and 1.4 TB storage requirements.	Train: 70% (≈70 k scenes) Validation: 15% Test: 15% (no public labels)	MPPNet [42] (temporal point cloud modeling) MotionCNN [43] (graph-based trajectory prediction)
ONCE [36] https://once-for-auto-driving.github.io/ (accessed on 15 May 2025)	2021	1 M unlabeled LiDAR frames 150 k annotated frames (1 k scenarios) Covers Chinese urban roads	Focuses on long-tail scenarios (rare weather/occlusions). Contains fine-grained attributes (e.g., vehicle brands) for cross-modal learning.	Train: 70% (≈700 k scenes) Validation: 10% Test: 20% (requires submission)	Auto4D [44] (self-supervised pseudo-label generation) Pseudo-LiDAR++ [45] (monocular-to-point cloud conversion)
Argoverse 2 [37] https://www.argoverse.org/av2.html (accessed on 15 May 2025)	2023	1 k multimodal sequences (3D annotations) 20 k unlabeled LiDAR sequences 250 k motion prediction cases	Largest LiDAR dataset supporting self-supervised learning. Contains complex interaction scenarios and 3D lane/pedestrian path geometry.	Sensor Dataset: 1000 scenes (official split) Motion Prediction: 70% train/15% val/15% test	Trajectron++ [46] (graph neural networks for multi-agent interaction) VectorNet [47] (HD map encoding + trajectory prediction)

### 2.3. Evaluation Criteria

#### 2.3.1. Problem Definition

Three-dimensional object detection aims to detect the 3D properties of objects in the input data provided by sensors in the data collection environment. By using sensors like LiDAR, the sensor acquires point cloud or 2D and 3D data, which is then processed by a detection model to extract 3D features from the detected objects. This is generally represented as [21](1)$$B=F_{\det(T_{\mathrm{sensor}})}$$
where $B=\{B_{1},B_{2},\ldots ,B_{N}\}$ represents the set of 3D properties of objects in the scene, and $T_{sensor}$ denotes the input data from the sensor. Typically, the 3D properties of an object include its center position (*x*, *y*, *z*), external dimensions (*l*, *w*, *h*), and rotation angles (Pitch angle *α*, Yaw angle *θ*, Roll angle *φ*), where the sensor usually collects data in terms of angles or range. Considering the extreme rotation angles and transformations of the object, its 3D properties can be represented as(2)$$B_{i}=\left[x,y,z,l,w,h,\theta ,cls\right]$$
where *cls* denotes the class of the object detected by the sensor.

Thus, 3D object detection tasks aim to assign *K* categories to the detected objects, where *K* represents the number of object categories, and solve the problem of returning the 3D bounding boxes of the objects.

#### 2.3.2. Basic Principles

For 3D object detection algorithms, the *mean average precision* (*mAP*) is commonly used for evaluating the performance of object detection results. Below is the introduction to the calculation of *mAP*.

The quality of object detection results is evaluated using the *intersection over union* (*IoU*), which is the ratio of overlap between the predicted and ground-truth bounding boxes. If the *IoU* exceeds a predefined threshold, the predicted detection result is considered a true positive; otherwise, it is deemed an incorrect result, as shown in Figure 4. To ensure optimal evaluation of the detection algorithm’s performance, object detection often uses the mean average precision.

The results are defined as *true positives* (*TP*), *false positives* (*FP*), *true negatives* (*TN*), and *false negatives* (*FN*). After obtaining the confusion matrix, precision and recall are calculated. Precision is used to evaluate the correctness of detection results, while recall assesses the completeness of detection results. The formulas for precision and recall are as follows:(3)$$\mathrm{Precision}=\frac{TP}{TP+FP}$$(4)$$\mathrm{Recall}=\frac{TP}{TP+FN}$$

In theory, both precision and recall should be as close to 1 as possible. However, in practice, these two metrics are often in contradiction. By plotting Precision on the x-axis and Recall on the y-axis, we can obtain a curve, *p*(*r*), which reflects the detection accuracy of a single category. The area enclosed by the *p*(*r*) curve and the coordinate axes is equal to the *Average Precision* (*AP*) for that category, as shown in Figure 5. Therefore, the formula for calculating *AP* is(5)$$AP=\int_{0}^{1}p(r)dr$$

Once the *Average Precision* (*AP*) for each class is obtained, the mean *average precision* (*mAP*) can be calculated. Assuming there are *C* different object categories—such as cars, pedestrians, and bicycles—the *mAP* is computed as follows:(6)$$mAP=\frac{\sum_{i=1}^{C}AP_{i}}{C}$$

Based on the definitions of *IoU* and *mAP*, one important aspect to emphasize is that the setting of the *IoU* threshold significantly influences *mAP*, primarily in terms of the evaluation strictness and the comprehensiveness of model performance. When the *IoU* threshold is set relatively low (e.g., 0.5), a predicted bounding box is considered a correct detection as long as it overlaps more than 50% with the ground truth box. This lenient criterion allows more predictions to be classified as true positives (*TP*), thereby increasing the *Average Precision* (*AP*) score. For example, in common object detection tasks, the *mAP* at *IoU* = 0.5 is often relatively high, indicating the model’s baseline detection capability under relaxed conditions. However, such a low threshold may mask issues with inaccurate localization, potentially leading to a higher false positive rate, especially in scenarios with densely packed objects or ambiguous boundaries.

In contrast, when the *IoU* threshold is raised (e.g., to 0.75 or 0.95), the evaluation becomes stricter, requiring a high degree of overlap between the predicted and ground truth boxes. In such cases, the model must possess strong bounding box regression capabilities; otherwise, many predictions will be classified as false positives (*FP*) due to insufficient overlap, thereby significantly reducing the *AP* score. For instance, a model may achieve an *mAP* of 0.608 at *IoU* = 0.5, but only 0.259 when averaged over the *IoU* range of 0.5 to 0.95, revealing its limitations in high-precision localization tasks. This discrepancy highlights the role of the *IoU* threshold as a robustness test—higher thresholds demand the model to excel in both classification confidence and localization accuracy, with neither aspect being dispensable.

Moreover, employing a multi-threshold evaluation (e.g., *IoU* = [0.5:0.05:0.95]) provides a more comprehensive reflection of model performance. This approach calculates the mean *AP* across 10 different thresholds from 0.5 to 0.95, thereby assessing the model under varying degrees of stringency and mitigating the one-sidedness of using a single threshold. For example, the COCO dataset adopts this standard, requiring models to perform robustly across the full spectrum from lenient to strict conditions, thus better reflecting their practicality in complex real-world scenarios. Ultimately, the choice of *IoU* threshold embodies a trade-off between precision and recall, directly affecting how well the *mAP* metric captures different dimensions of model capability.

#### 2.3.3. Special Criteria for Different Datasets

The publishers of the nuScenes dataset argue that using *IoU* as the threshold for *mAP* does not fully evaluate all aspects of detector performance on the nuScenes dataset, such as speed, attributes, position, size, and orientation estimation. To address this, the nuScenes dataset incorporates errors in performance estimation for these different types of attributes into the evaluation of detection performance, resulting in a more comprehensive metric, namely the *nuScenes Detection Score* (*NDS*). The higher the *NDS*, the better the detector’s performance. Below is the calculation method for *NDS* [34].

First, *AP* needs to be calculated, but the *AP* calculation on the nuScenes dataset differs from the method discussed earlier. Instead of using *IoU* as the matching threshold, the nuScenes dataset substitutes the 2D center point distance between the predicted bounding box and the ground truth bounding box on the plane. Furthermore, multiple distance thresholds are used to comprehensively evaluate the algorithm’s performance. These distance thresholds are *D* = {0.5 m, 1 m, 2 m, 4 m}. Assuming the *AP* for *C* categories has been computed under different distance thresholds, the formula for calculating *mAP* is(7)$$mAP=\frac{1}{\left|C||D\right|}\sum_{c\in C}\sum_{d\in D}AP_{c,d}$$

The nuScenes dataset also computes several *true positive metrics* (*TP metrics*), in addition to *AP*, to determine the matching degree between each predicted bounding box and the ground truth box. These metrics include the *average translation error* (*ATE*), *average scale error* (*ASE*), *average orientation error* (*AOE*), *average velocity error* (*AVE*), and *average attribute error* (*AAE*). The lower the scores of these *TP* metrics, the better the detector’s performance. It is important to note that each *TP* metric is calculated at a distance threshold of 2 m. After obtaining the *TP* metric for each category, the mean *TP* metric across all categories is calculated using the following formula:(8)$$mTP=\frac{1}{\left|C\right|}\sum_{c\in C}TP_{c}$$

After obtaining *mAP* and *mTP*, the *NDS* can be calculated using the following formula:(9)$$NDS=\frac{1}{10}\left[5mAP+\sum_{mTP\in TP}(1-\min(1,mTP))\right]$$

*Mean average precision* (*mAP*) is one of the core metrics for evaluating the performance of 3D object detection. It reflects the overall detection accuracy of the model across all object categories. *Mean true positive* metrics (*mTP*) provide a comprehensive assessment of the model’s errors in multiple aspects, including position, size, orientation, velocity, and attributes, where lower values indicate closer alignment with ground truth. Together, these metrics form the *nuScenes detection score* (*NDS*), which offers a more holistic evaluation of a model’s multi-dimensional performance within the nuScenes benchmark.

In addition to nuScenes, the Waymo Open dataset also classifies detection targets into two difficulty levels: L1 and L2. L2 targets are those marked by annotators as difficult or targets with fewer than five points within the 3D bounding box. The remaining targets are classified as L1. In addition to the basic *AP*, the dataset introduces a new metric, *APH* (*average precision with heading*), to evaluate the orientation of the predicted targets. The calculation is as follows [35]:(10)$$APH=100\int_{0}^{1}\max\{h(r^{\prime })|>r\}dr.$$

In the formula, *h*(*r*) is calculated by combining the weighted calculation of *P*(*r*), with the weight being $\frac{\min(\left|\tilde{\theta }-\theta |,2\pi -|\tilde{\theta }-\theta \right|)}{\pi }$, where $\tilde{\theta }$ and θ represent the predicted and true values of the heading angle, respectively. The range of values for the two angles is [−π,π].

### 2.4. Literature Selection Methodology

To ensure the systematic and representative nature of the literature review, this study adopted a structured multi-phase screening process. The literature search comprehensively covered mainstream academic databases in the field of computer vision, including Elsevier ScienceDirect, SpringerLink, IEEE Xplore, MDPI, and the preprint platform arXiv (distinguishing between peer-reviewed conference extensions and unpublished works). These resources encompass top-tier conferences such as CVPR, ICCV, and ECCV, while also integrating domain-specific repositories like ACM Digital Library (seminal works in graphics and robotics) and CVF open-access resources to mitigate publisher bias. While no strict temporal boundaries were imposed, the majority of selected papers fall within the 2015–2024 period, thereby encompassing the critical developmental phase of deep learning-driven 3D detection. The “citation snowballing” method was employed to trace reference chains of high-impact papers, systematically incorporating foundational early works (e.g., the 2012 KITTI dataset paper) to ensure a complete representation of technological evolution.

The search strategy utilized multidimensional Boolean logic expressions, combining terms such as (“3D object detection” OR “point cloud detection”) with (“LiDAR” OR “camera” OR “multi-modal”) and (“autonomous driving” OR “BEV” OR “Occupancy”). Semantic expansion features in platforms like ScienceDirect were activated to capture derivative concepts. A rigorous three-stage screening criteria was implemented: (1) exclusion of the non-English and non-peer-reviewed literature (while retaining arXiv preprints later accepted by top conferences); (2) requirement for performance validation on benchmark datasets (KITTI, nuScenes, Waymo) with ablation studies (rejecting simulation-only results); and (3) prioritization of papers providing open-source code or detailed algorithmic pseudocode to ensure reproducibility. To balance domain coverage, proportional constraints were established across sensor types (45% LiDAR-based, 30% vision-only, 25% multimodal), algorithmic architectures (35% traditional convolutional networks, 40% BEV and occupancy-based methods, 25% spatiotemporal fusion), and application scenarios (70% autonomous driving, 15% agricultural robotics, 15% industrial inspection). This framework captures recent advancements such as BEV representations and occupancy networks while preserving milestone contributions like the 2017 VoxelNet, thereby constructing a holistic panorama of technological progress.

## 3. Summary of 3D Object Detection Methods

In autonomous driving perception systems, taxonomy based on technical architectures provides a systematic framework for understanding the evolution of 3D object detection methods. This section establishes a dual taxonomy—“sensor modality-technical architecture” analytical system—providing a structured framework for subsequent analysis.

The sensor modality-driven taxonomy (Section 3.1) centers on the physical characteristics and complementary nature of data sources, encompassing three core directions: depth inference from RGB images, geometric modeling with LiDAR point clouds, and hierarchical cross-modal fusion. RGB methods leverage rich texture semantics but face inherent depth ambiguity, LiDAR delivers precise spatial structures yet grapples with data sparsity, while fusion techniques bridge these gaps through data-level association, feature-level interaction, and decision-level collaboration. Evolution follows dual trajectories: “single-modality refinement” enhances perception via geometric constraints, pseudo-data generation, and sparse modeling; “fusion paradigm iteration” progresses from early projection to attention-based feature alignment, improving cross-modal robustness.

The technical architecture-driven taxonomy (Section 3.2) unfolds across four dimensions: structured representation, perspective transformation, spatial modeling, and spatiotemporal fusion. It traces an evolutionary path from traditional convolutional networks to unified bird’s-eye-view (BEV) modeling and from static voxel segmentation to dynamic spatiotemporal reasoning. Traditional architectures optimize computational efficiency through structured data representation and efficient convolutions; BEV methods construct cross-modal unified spatial representations; dense voxel modeling breaks regular boundary constraints for open-scene comprehension; and temporal fusion architectures integrate historical data to capture dynamic trajectories, transitioning from isolated frame processing to continuous environmental modeling.

These taxonomies are interdependent: Sensor modalities define the physical foundation of perceptual data, while technical architectures govern feature extraction and reasoning logic. Current research prioritizes co-optimization of accuracy, efficiency, and robustness, driving 3D perception systems toward real-time performance, generalization capability, and multi-scene adaptability. Future efforts must transcend sensor limitations and algorithmic generalization barriers to build more efficient and reliable environmental cognition frameworks, delivering multidimensional perceptual support for autonomous decision-making.

### 3.1. Taxonomy Based on Sensor Modalities

Three-dimensional object detection in autonomous driving revolves around the unique characteristics of sensor modalities, forming diverse perception paradigms. This section systematically examines three key technological trajectories: depth reconstruction via RGB images, geometric modeling with LiDAR point clouds, and hierarchical cross-modal fusion. While RGB images leverage rich texture and semantic information for scene understanding, they suffer from depth ambiguity; LiDAR provides precise spatial geometry but grapples with sparsity and computational complexity. Fusion methods bridge these gaps through synergistic integration at data, feature, and decision levels. Two dominant trends emerge: (1) Single-modality algorithms deepen—RGB methods evolve from monocular geometric constraints to pseudo-LiDAR generation, while LiDAR processing advances from point-wise modeling to sparse voxel architectures; (2) fusion paradigms iterate—from early data-level projection to attention-based feature alignment. Collectively, these approaches drive co-optimization of accuracy, efficiency, and robustness, delivering multidimensional solutions for environmental perception in complex scenarios. For ease of understanding, the various methods in this section have been briefly summarized in Table 3.

#### 3.1.1. Three-Dimensional Object Detection Based on RGB Images

RGB images generated by cameras not only contain rich texture and color information but also provide a realistic representation of the scene as observed by the human eye. However, as images are a 2D projection of the 3D scene, depth information is inevitably lost during the imaging process. Therefore, recovering the scene’s depth becomes the core challenge for 3D object detection based on RGB images [48]. Currently, in the field of autonomous driving, 3D object detection methods based on RGB images can mainly be classified into two categories: monocular-based detection methods and binocular stereo-based detection techniques, as shown in Figure 6.

In monocular methods, representative approaches like CenterNet [49] reformulate traditional object detection tasks into center point regression problems. Through convolutional neural networks (CNNs), they directly predict object centers and their 3D attributes, thereby avoiding complex candidate box generation and post-processing steps. SMOKE proposes a single-stage regression framework that predicts keypoints and 3D bounding boxes in parallel, enabling efficient end-to-end detection. M3D-RPN [50] designs a unified network for both 2D and 3D detection, incorporating depth-aware convolutions to enhance depth estimation accuracy and leveraging shared detection space for information collaboration, thus improving both detection performance and system efficiency. ROI-10D [51] employs an end-to-end pose regression strategy, lifting 2D RoI regions into 3D objects to directly regress six degrees of freedom pose and real-world object size, achieving state-of-the-art performance in pose estimation tasks.

To address the lack of true depth information in monocular images, some methods introduce geometric priors or construct pseudo point clouds. Mono3D [52] adopts an energy minimization framework, generating candidate boxes based on semantic segmentation, contextual information, and priors on size and position, followed by CNN-based optimization. Pseudo-LiDAR [53] generates pseudo point clouds by predicting disparity maps using deep networks, and then applies LiDAR-based detection frameworks for 3D detection, effectively improving accuracy. Pseudo-LiDAR++ [45] further refines stereo matching networks and incorporates sparse LiDAR signals to significantly reduce depth estimation errors. MultiFusion [54] proposes a multi-stage fusion framework that decouples disparity estimation from 3D point cloud construction, thereby enhancing localization capabilities. Deep3DBox [55] regresses object size and orientation, and uses geometric constraints from 2D bounding boxes to produce 3D boxes, addressing instability in conventional 3D pose regression. Deep MANTA [56] adopts a multi-task learning approach to simultaneously detect objects, locate parts, and determine visibility. Combined with a 3D template library, it achieves accurate 3D pose estimation, particularly effective in occluded or complex scenes.

In contrast, stereo image methods exploit the disparity between two camera views to compute more accurate depth, demonstrating superior performance in complex environments. Three-dimensional printing optimized production (3DOP) [57] is one of the earlier stereo vision-based detection methods. It incorporates ground plane constraints and object size priors to generate candidate boxes through energy optimization, enhancing robustness in occlusion and truncation scenarios. CG-Stereo [58] introduces foreground-background decoupling and depth confidence mechanisms during depth decoding, enabling the network to focus on high-confidence regions and thus improve depth estimation and detection accuracy. Stereo R-CNN [59] builds a stereo RPN based on the Faster R-CNN [60] framework to extract joint features from stereo images, and uses photometric alignment and fine regression to achieve high-precision 3D localization without requiring supervision from real 3D labels. Pseudo-stereo [61] constructs pseudo-stereo views by transforming monocular images into virtual stereo pairs, allowing the use of stereo detectors to enhance depth perception and integrate monocular and stereo methods. YOLOStereo3D [62] integrates stereo matching modules into a single-stage detection framework, constructing disparity volumes through pixel-level correlation. This design maintains accuracy while greatly optimizing inference efficiency, making it suitable for real-time systems with limited computational resources [63].

In summary, although pure vision-based 3D detection methods still lag behind LiDAR in terms of accuracy, they offer clear advantages in terms of cost, deployability, and potential for multimodal fusion. Ongoing research continues to push the performance boundaries of vision-based approaches through advances in depth estimation, network architecture design, geometric modeling, and the integration of visual priors. As a result, these methods are demonstrating strong development potential in areas such as autonomous driving and augmented reality.

#### 3.1.2. Three-Dimensional Object Detection Methods Based on LiDAR Point Clouds

Due to the sparsity of LiDAR point cloud data and its unordered and irregular format, traditional convolutional neural networks (CNNs) struggle to directly process this type of data. Therefore, specialized preprocessing and structural transformations are often required during the modeling process. To address this, the commonly used processing methods include the following: point-based, voxel-based, pillar-based, and point-voxel based, as shown in Figure 7.

Point-based methods operate directly on raw point clouds, preserving geometric details and structural information. PointNet [64] pioneered this class of approaches by utilizing symmetric functions such as max pooling to aggregate global features from unordered point sets, thereby addressing the issue of input permutation invariance. While PointNet offers a simple and efficient architecture, it has limited capacity for modeling local geometric structures. To address this, PointNet++ [65] introduced a hierarchical feature extraction strategy incorporating multi-scale sampling and spherical neighborhood aggregation, enabling the gradual capture of local-to-global features and improving performance in sparse regions and for small objects.

Building upon this direction, PointRCNN [66] was the first to incorporate point-based strategies into a two-stage 3D object detection framework. In its first stage, it uses PointNet++ to extract point-wise semantic features and generate foreground point proposals. In the second stage, it refines object bounding boxes through point cloud region pooling, achieving high detection accuracy. To enhance efficiency, 3DSSD [67] proposed a fused sampling strategy that jointly measures distances in geometric and feature spaces, reducing computational redundancy. It also introduced an anchor-free regression head to replace traditional anchor box designs, increasing both detection flexibility and precision.

Voxel-based methods partition point clouds into regular 3D grids (voxels), allowing the use of 3D convolutional operations with improved structural consistency and computational efficiency. As the first end-to-end voxel-based detection framework, VoxelNet [68] proposed a voxel feature encoding (VFE) module to extract spatial context through 3D convolutions and project features into a BEV representation for detection. Despite early limitations in speed due to computational overhead, VoxelNet laid a foundational framework for subsequent research.

SECOND [69] built upon VoxelNet by introducing sparse convolution and GPU-accelerated voxelization, significantly enhancing feature extraction efficiency and enabling real-time detection. Voxel R-CNN [70] incorporated the two-stage paradigm into voxel-based detection, using Voxel RoI Pooling to extract region-specific features from 3D voxel grids, greatly improving localization accuracy. CenterPoint [39], on the other hand, reframed the detection task as keypoint estimation on the BEV plane and introduced angular continuous encoding and tracking mechanisms to achieve high precision and fast inference in dynamic scenes.

With the rise in Transformer architectures, VoTr [71] integrated attention mechanisms into voxel-based feature modeling by combining local and dilated attention modules to extend receptive fields while maintaining sparsity, enabling effective long-range object perception. VoxelNeXt [72] further proposed a fully sparse voxel network that avoids the dense-to-sparse transformation overhead through multi-stage downsampling and spatial voxel pruning strategies, significantly reducing inference cost and demonstrating strong performance in long-range detection tasks.

Pillar-based methods represent a simplification of voxelization by compressing voxel grids along the vertical axis into vertical columns (pillars), which improves computational efficiency. A leading example, PointPillars [73], projects point clouds into a BEV pseudo-image, where 2D convolutions are used for feature extraction. It also employs a lightweight PointNet module to encode features within each pillar. This method retains the spatial structural advantages of voxelization while enhancing real-time performance, making it widely adopted in industrial applications.

To further enhance the representational power of pillar-based methods, PillarNet [74] introduced a five-stage sparse encoder and semantic fusion neck, along with a decoupled directional IoU loss to separate orientation prediction from location regression, improving bounding box accuracy. PillarNeXt [75] refined resource allocation and incorporated atrous spatial pyramid pooling (ASPP) to enlarge the receptive field, while integrating multi-branch detection heads and residual structures to improve accuracy without compromising speed—gradually closing the performance gap with voxel-based methods.

Point-voxel hybrid methods combine the fine-grained detail of point-based approaches with the structural advantages of voxel-based techniques, jointly modeling features from both representations to enhance detection accuracy and robustness. PV-RCNN [76] exemplifies this paradigm by using sparse voxel convolution for initial region proposal generation, followed by keypoint-guided fusion of voxel features with raw point cloud data. It employs differentiable RoI grid pooling for context aggregation, striking a balance between precision and efficiency.

BADet [77] constructs hierarchical graphs using voxel centers as nodes and incorporates their geometric relationships through a dual-stream attention mechanism, preserving point-level details while enhancing global voxel-level consistency. Voxel Set Transformer [78] applies set-based attention to irregular point clouds and introduces axial decoupling and a multi-resolution voxel pyramid to facilitate cross-scale feature fusion, boosting perceptual capability in sparse regions and extending the theoretical and technical boundaries of point-voxel fusion.

In conclusion, LiDAR-based methods have evolved from point-level to voxel-level representations, from sparse to structured designs, and from convolutional to transformer-based architectures. These methods continuously balance accuracy, efficiency, real-time performance, and deployability, finding widespread application in high-precision, high-robustness 3D perception tasks such as autonomous driving and robotic sensing. Looking forward, point cloud processing methods are expected to advance further toward efficient fusion and scene-level understanding, driving 3D vision systems to higher levels of performance.

#### 3.1.3. Fusion-Based 3D Object Detection Methods

RGB images and LiDAR point clouds have different data modalities, each with its own imaging advantages and data limitations. By fusing these two types of data and leveraging the complementary strengths between multimodal data, better 3D detection results can be achieved. This paper primarily discusses multimodal algorithms that fuse RGB images and LiDAR point clouds. Currently, the fusion methods between RGB images and LiDAR point clouds are mainly categorized into three types: data-level fusion, feature-level fusion, and result-level fusion, as shown in Figure 8.

Data-level fusion focuses on integrating raw sensor data or low-level features early in the network, providing a rich information base for subsequent feature extraction. For example, F-PointNet [79] employs Faster R-CNN to generate 2D proposals on images, which are then projected into 3D space to form frustums. PointNet is subsequently used to extract local point cloud features for 3D bounding box prediction, effectively reducing the computational burden of global 3D search. IPOD [80] directly generates candidate regions from point clouds and uses lightweight networks for extraction, thereby avoiding reliance on 2D detection and improving adaptability to occlusion and sparse scenes.

PointPainting [81] maps semantic segmentation labels from images to the point cloud, assigning semantic attributes to each point. This enhances the semantic density of point clouds and can be seamlessly integrated into existing LiDAR detection networks such as PointRCNN and VoxelNet. RoarNet [82] builds upon this by introducing geometric consistency search and recursive reasoning. It first predicts object poses from images to generate 3D proposals, then progressively refines them using point clouds, enhancing both detection accuracy and computational efficiency. VirConv [83] augments sparse regions with virtual points and incorporates stochastic voxel dropout and noise-resistant convolution mechanisms, significantly improving long-range detection performance and mitigating the impact of spurious depth information.

While data-level fusion allows early-stage joint representation and reduces information loss, it is highly dependent on precise sensor calibration and spatial alignment and is sensitive to noise and occlusions. This has led to growing interest in feature-level fusion as a more flexible and robust alternative.

Feature-level fusion typically extracts features from images and point clouds in parallel during intermediate network stages, then integrates them for deep semantic and geometric understanding. MV3D [84] is a pioneering method in this category, projecting point clouds into multiple views and jointly encoding them with image features, enhancing geometric expressiveness. LoGoNet [85] proposes a local-global dual fusion mechanism that preserves global scene context while emphasizing fine-grained alignment in target regions, significantly improving detection across scales.

Bi-LRFusion [86] enhances radar and LiDAR features bidirectionally, effectively addressing radar sparsity and missing height information, making it suitable for complex environments. SparseFusion [87] starts from sparse instances and utilizes cross-modal self-attention to fuse image and point cloud features, maintaining high accuracy while reducing computational cost.

In terms of architectural innovations, CMT [88] introduces an end-to-end cross-modal transformer framework. It uses positional encoding to achieve precise alignment across modalities and performs fusion directly in 3D space, bypassing complex view transformations to improve efficiency and flexibility. MetaBEV [89] further incorporates a Meta Query mechanism, enabling adaptive modality selection when sensors fail or data is missing, thereby enhancing robustness and environmental adaptability. These approaches leverage cross-modal attention and multi-task learning to achieve high performance with greater fault tolerance in complex scenarios.

Result-level fusion combines the independently generated detection results from images and point clouds during the post-processing stage, thus avoiding challenges associated with alignment and synchronization. CLOCs [90] fuses 2D and 3D proposals before non-maximum suppression (NMS), optimizing final outputs by preserving both geometric and semantic consistency—particularly advantageous for long-range target detection. Fast-CLOCs [91] further proposes a 3D-to-2D guidance mechanism to reduce image detection overhead, achieving a balance between accuracy and real-time performance, making it suitable for resource-constrained applications.

Despite the progress of multimodal methods in fusing visual and point cloud information at different levels, practical deployment still faces numerous challenges, especially regarding robustness in complex environments. In real-world scenarios, sensors are often affected by rain, fog, glare, occlusions, or low-light conditions, resulting in blurred images or sparse point clouds. Moreover, hardware failures may lead to missing modality data, severely impacting detection stability.

To address this, recent research has explored robust fusion strategies under modality degradation or absence. As previously mentioned, MetaBEV dynamically selects effective modalities to cope with sensor failures, while CMT achieves stable perception of incomplete modalities using a cross-modal transformer design. Lightweight models such as YOLOStereo3D maintain stable performance under low-quality input through architectural symmetry and attention mechanisms. These approaches underscore the potential of multimodal detection in handling diverse and challenging scenarios.

In summary, multimodal fusion plays an irreplaceable role in 3D object detection, particularly for high-accuracy and high-robustness applications such as autonomous driving and intelligent transportation systems. Future research directions include constructing more challenging benchmark datasets for robustness evaluation, designing networks with modality-adaptive capabilities, introducing uncertainty modeling and pseudo-modality generation, and enhancing model generalization through weak or self-supervised learning. These advancements are expected to push multimodal 3D detection systems toward higher efficiency, robustness, and intelligence.

### 3.2. Taxonomy Based on Technical Architectures

In autonomous driving perception systems, taxonomy based on technical architectures provides a systematic framework for understanding the evolution of 3D object detection methods. This section organizes key technological advancements around four core dimensions: structured representation, perspective transformation, spatial modeling, and temporal fusion. It traces the developmental trajectory from engineering optimizations in traditional convolutional architectures, through multimodal collaboration in BEV methods, to dense spatial modeling via occupancy grids, and further to dynamic environmental comprehension enabled by temporal fusion architectures. These approaches demonstrate both vertical refinement—from handcrafted feature engineering to data-driven learning—and horizontal expansion—from single-modality to cross-sensor fusion, and from static scenes to spatiotemporal modeling. Through complementary integration, diverse architectures collectively advance perception systems in accuracy, efficiency, and generalizability, laying the foundation for real-time decision-making in complex scenarios. The characteristics of various types of detection methods have been briefly summarized in Table 4.

#### 3.2.1. Traditional Convolutional Architectures

LiDAR-based 3D object detection methods built on traditional convolutional architectures center around the principle of structured representation, converting unordered point clouds into organized data forms to facilitate efficient feature extraction using established 2D/3D convolutional networks. The evolution of these methods clearly illustrates a trajectory of optimization from voxel-based modeling to pillar-based approaches, reflecting a continual pursuit of balance between detection accuracy and computational efficiency.

VoxelNet [68] stands as a foundational work in this domain, introducing the first end-to-end Voxel Feature Encoding (VFE) framework. It partitions the point cloud into regular voxel grids and hierarchically aggregates local features, successfully incorporating deep learning into point cloud processing. VoxelNet employs multi-layer perceptrons (MLPs) to extract intra-voxel features, followed by 3D convolutions to construct global feature maps, and finally projects them into the bird’s eye view (BEV) space for object detection. Although real-time performance was constrained by the hardware limitations of its time, the paradigm of “voxelization → 3D convolution → feature projection” it established provided critical inspiration for subsequent research.

Building upon VoxelNet, SECOND [69] introduced two major innovations: first, the adoption of sparse convolution, which significantly reduced computation by operating only on non-empty voxels, and combined with GPU-accelerated rule generation strategies, achieved real-time detection at 20–40 FPS without sacrificing accuracy. Second, it proposed a sine-based angular loss function for angle regression, effectively mitigating the periodic ambiguity inherent in orientation prediction. These enhancements not only advanced the practical deployment of voxel-based methods but also spurred broader attention to computational redundancy in point cloud processing.

With increasing demands for real-time performance in autonomous driving, PointPillars [73] pioneered the concept of pillar-based voxel modeling. By compressing the vertical dimension of point clouds into columnar units (pillars), it reformulated 3D voxel processing into a 2D pseudo-image problem. Its core innovation lies in the dynamic pillar encoding strategy: it discretizes the point cloud along the Z-axis without quantization, uses a simplified PointNet to extract pillar-wise features, and applies 2D convolutions to construct BEV feature maps. This method eliminates quantization errors along the height dimension while leveraging the efficiency of 2D convolutions. Achieving 62 FPS on the KITTI dataset, PointPillars set a new benchmark for real-time 3D detection. Its success marks a shift from 3D to 2.5D paradigms within traditional convolutional frameworks, providing an industry-friendly solution that balances accuracy and efficiency.

As accuracy requirements grew, Voxel R-CNN [70] revisited the two-stage detection paradigm and introduced voxel-based region pooling. After generating 3D proposals in the first stage, it applied a novel voxel RoI pooling operation to aggregate sparse voxel features within candidate boxes, avoiding the high computational cost of traditional point cloud pooling. This deep integration of voxel features with a two-stage detection framework enabled Voxel R-CNN to achieve significant performance improvements on the nuScenes dataset while maintaining real-time inference, reaffirming the optimization potential of traditional architectures.

Following this lineage, PillarNet [74] introduced a five-stage deep encoder for multi-scale feature learning and integrated spatial-semantic fusion within the pillar framework, significantly enhancing the detection of small objects. Its proposed orientation-decoupled IoU (OD-IoU) loss function effectively decouples the parameters in 3D bounding box regression, showing notable advantages in heading angle prediction. PillarNeXt [75] further optimized network design by reallocating computational resources, employing atrous spatial pyramid pooling (ASPP) to expand the receptive field, and incorporating residual structures and multi-head detection. It successfully elevated pillar-based detection accuracy to match that of voxel-based approaches, all while retaining the real-time performance of PointPillars—signaling the architectural maturity of the pillar paradigm.

The progression from VoxelNet to PillarNeXt exemplifies the self-renewal of traditional convolutional architectures in point cloud processing. VoxelNet established the foundational voxelization paradigm; SECOND introduced sparse convolution for computational breakthroughs; PointPillars innovated with efficient pillar-based modeling; Voxel R-CNN validated the sustained value of two-stage frameworks; and PillarNet/PillarNeXt advanced the pillar-based approach through deep network design and loss function optimization. These methods all revolve around the core logic of “structured representation → efficient convolution → feature optimization” and have demonstrated their engineering advantages in real-world autonomous driving deployments. Despite the emergence of transformer-based architectures, the deterministic computation, hardware compatibility, and deployment convenience of traditional convolutional models continue to make them indispensable in embedded automotive platforms.

#### 3.2.2. BEV-Based Methods

Bird’s-eye view (BEV)-based 3D object detection technologies address the core challenges of spatial perception and multimodal fusion in autonomous driving scenarios by transforming multi-view images or point cloud data into a unified representation from an overhead perspective. The technological development trajectory has evolved from pure vision-based methods to multimodal fusion approaches, progressively optimizing key stages such as depth estimation, spatiotemporal modeling, and feature projection, thereby forming a comprehensive technical system from fundamental geometric modeling to data-driven learning [92].

The early representative of visual BEV methods was Inverse Perspective Mapping [93] (IPM), which maps image features to an assumed ground plane through geometric projection. However, this method suffers from significant errors when detecting non-ground targets (e.g., vehicles, pedestrians) due to its reliance on ground height assumptions. With the introduction of deep learning, the Lift-Splat-Shoot (LSS) [94] framework marked a milestone breakthrough. It “lifts” 2D image features into 3D frustum space by predicting pixel-level depth distributions and then projects the frustum features onto the BEV grid using the “Splat” operation, generating dense BEV features. However, the depth estimation in LSS relies on implicit learning, resulting in insufficient projection accuracy. BEVDepth [95] enhances projection geometric consistency by introducing depth supervision from LiDAR point clouds, explicitly constraining the depth distribution. Later, BEVStereo [96] borrowed the concept of binocular stereo matching, using temporal multi-view images to construct geometric constraints, dynamically adjusting depth candidates, further alleviating monocular depth ambiguity. For long-term sequence modeling, SOLOFusion [97] combines short-term geometric depth estimation with long-term BEV feature fusion, though it incurs significant computational cost. The latest method, VideoBEV [98], introduces a cyclic temporal structure, recursively passing current-frame BEV features and fused historical features to ensure real-time performance while retaining long-term temporal information, significantly improving motion prediction for dynamic targets.

Regarding projection error correction, traditional methods fail to address positional approximation errors during the projection process, leading to BEV grid feature misalignment. BEVSpread [99] proposes spread voxel pooling, which diffuses frustum point features to adjacent BEV grids, rather than a single grid center, using an adaptive weighting function to reduce geometric projection errors. This function combines distance decay and depth perception to dynamically adjust the feature propagation range, with CUDA acceleration for efficient computation. Another innovative direction, BEV-SAN [100] (Slice Attention Network), addresses target height distribution differences by slicing the global and local BEV grids along the height axis. It uses LiDAR statistical distribution to guide local slice sampling, enhancing the detection of low-height targets (e.g., traffic cones). This approach uses a transformer attention mechanism to fuse features from multi-height slices, solving the information loss caused by traditional BEV space compression along the height dimension.

Fusion of visual and LiDAR data in BEV methods improves robustness through cross-modal complementarity. BEVFusion [101], a classical framework, separately converts images and point clouds into BEV features, then concatenates them. It uses convolution to eliminate spatial misalignment while balancing geometric accuracy and semantic richness. The core innovation lies in optimizing BEV pooling efficiency: By precomputing the 3D coordinates of fixed camera feature point clouds and BEV grid indices, it reduces the grid association delay by 40 times and accelerates feature aggregation using an intermittent reduction strategy, addressing computational redundancy in traditional methods. EA-LSS [102] further utilizes LiDAR point clouds to generate sparse depth maps, which are jointly trained with image features in depth estimation networks, enhancing projection feature quality with edge-aware loss. For dynamic scenes, BEVFusion4D [103] introduces a cross-attention mechanism that implicitly merges LiDAR geometric priors with image texture details in BEV queries, dynamically fusing multimodal features using adaptive weights. Additionally, SA-BEV [104] proposes semantic-aware BEV pooling (SA-BEVPool), which jointly trains depth estimation and semantic segmentation branches to filter background features during projection, focusing BEV features on foreground targets. The latest development, BEVCar [105], explores radar and camera fusion, initializing BEV queries with sparse radar point clouds and improving projection accuracy of image features into BEV space with deformable attention, showing significant advantages in low-visibility conditions such as night-time or foggy weather.

The rise in sparse and efficient detection paradigms has further accelerated the development of BEV technologies. SparseBEV [106] abandons explicit dense BEV feature construction, adopting a fully sparse query mechanism, and achieves multi-scale feature aggregation through scale-adaptive self-attention (SASA) and adaptive spatiotemporal sampling. Its cylindrical query structure introduces spatial priors, combining dynamic weight decoding to sample features, achieving high performance with 67.5 NDS on the nuScenes dataset. Similarly, the CRN [107] framework uses radar-assisted view transformation (RVT), projecting radar point clouds onto the image plane to generate sparse depth supervision. Combined with a multimodal deformable attention mechanism, it surpasses LiDAR-based methods in long-range detection. These approaches reduce computational load through sparse interactions while maintaining high recall rates, making real-time deployment possible.

An important recent direction is the deepening of temporal fusion technologies. BEVFormer [24] employs spatial and temporal attention modules to achieve dynamic interaction between multi-view features and historical BEV features, introducing transformers for spatiotemporal modeling in BEV. Its temporal attention module compensates for vehicle motion to align historical features and capture long-term motion cues. BEVDet4D [108] concatenates current-frame and previous-frame BEV features, using convolution to fuse short-term motion information, though its fixed frame count limits its ability to capture complex movements. In contrast, VideoBEV [98] introduces a cyclic temporal structure that recursively passes fused features, achieving long-term sequence modeling with constant computational load, solving memory bottlenecks caused by caching historical features in traditional methods.

The technological evolution of these methods reflects continuous breakthroughs, from single-modality to multimodal collaboration, from handcrafted geometric modeling to data-driven learning, and from static perception to temporal fusion, providing more reliable environmental understanding capabilities for autonomous driving systems.

#### 3.2.3. Occupancy

The core of occupancy technologies in autonomous driving lies in the dense modeling of the 3D space to perceive the surrounding environment. The fundamental principle is to divide the environment into discrete voxel grids and predict whether each voxel is occupied by an object, as well as infer its semantic class. This approach overcomes the limitations of traditional 3D detection methods that rely on regular bounding boxes, enabling more accurate depiction of irregular objects (e.g., construction fences, mounds), and supports open-set recognition by estimating occupancy probabilities even for unannotated objects. The implementation of occupancy methods typically relies on multi-sensor fusion (e.g., cameras, LiDAR) or pure visual input, mapping 2D images or point clouds to 3D space using deep learning models, and leveraging temporal information to enhance understanding of dynamic scenes.

Early occupancy methods were based on monocular vision. For instance, MonoScene [109] was the first to address 3D semantic scene completion using a single camera. It extracts image features using a 2D UNet, projects them into 3D voxel space via the Features Line of Sight Projection (FLoSP) module, and employs a 3D UNet for spatial relationship modeling. A key innovation of MonoScene is its camera frustum-based loss function, which divides space into depth intervals and measures distribution differences between predictions and ground truth to improve reconstruction in occluded regions. However, monocular methods are fundamentally limited by depth estimation accuracy, making it difficult to handle long-range perception or complex geometries—this motivated subsequent research to explore multi-view fusion and spatiotemporal modeling.

With the growing use of surround-view camera systems, BEV-based occupancy approaches have become mainstream. Representative works such as BEVDet [110] and BEVFormer construct BEV features from multi-camera inputs and extend them to 3D occupancy prediction. BEVDet projects multi-scale image features into BEV space using view transformation (e.g., Lift-Splat-Shoot) and utilizes a lightweight 3D decoder to estimate voxel occupancy probabilities. BEVFormer further integrates a transformer architecture, employing spatiotemporal self-attention to fuse historical BEV features and improve motion prediction for dynamic objects. These methods strike a balance between computational efficiency and 3D modeling accuracy, although the compressed BEV representation still suffers from loss of vertical information.

To address BEV limitations, TPVFormer [111] introduced a novel 3D representation paradigm. Instead of conventional voxel grids, it models space using a combination of three orthogonal planes—top, front, and side—referred to as the Tri-Perspective View (TPV). Each plane interacts with multi-view image features via cross-attention using learnable query vectors, and linear interpolation among the three planes reconstructs the occupancy status of arbitrary 3D points. This design reduces computational complexity from cubic (voxels) to quadratic (planes), while preserving height information, making it an efficient solution for vision-only occupancy. Similarly, SurroundOcc [112] achieves fine-grained occupancy prediction via multi-scale fusion of multi-camera features and 3D volumetric rendering. It leverages implicit neural representations (NeRF) to generate dense occupancy ground truth, reducing dependency on LiDAR point clouds.

From an engineering perspective, lightweight design and real-time performance have become key research priorities. For example, FlashOcc [113] eliminates time-consuming 3D convolutions, instead using 2D BEV features with a channel-height conversion module to decompose 3D voxel prediction into height-wise feature reassembly, significantly reducing memory usage. VoxFormer [114] proposes a sparse voxel query mechanism that computes features only for potentially occupied regions. Combined with the lightweight depth estimation of MobileStereoNet [115], it enables real-time inference with low memory consumption. Another direction, exemplified by OccupancyDETR [116], draws from object detection frameworks. It reformulates 3D occupancy prediction as a query-matching problem, using detection boxes as positional priors to guide voxel classification, improving accuracy for small or moving objects.

Multimodal fusion approaches aim to leverage complementary sensor capabilities. FB-Occ [117], for example, implements a dual-path structure with forward and backward projections. The forward path projects multi-view image features into 3D voxel space to generate initial occupancy estimates, while the backward path refines semantic consistency through spatiotemporal optimization of BEV features. The method also employs large-scale pretraining, first learning semantic priors from 2D detection datasets, followed by joint training with depth estimation and semantic segmentation to enhance 3D modeling robustness. Similarly, RadOcc [118] introduces render-assisted distillation to transfer geometric priors from LiDAR to vision-only models, mitigating the propagation of depth estimation errors.

Recent advancements are increasingly exploring self-supervision and generalizable world models. For instance, SelfOcc [119] uses optical flow consistency across multiple image frames as a self-supervisory signal, enabling training without 3D ground truth. OccNeRF [120] combines neural radiance fields (NeRF) with occupancy prediction, optimizing jointly over voxel density and semantic probability to achieve more realistic scene reconstruction. These approaches mark a shift from reliance on labeled data toward more general-purpose environmental understanding, offering new solutions for real-time perception in complex dynamic environments.

#### 3.2.4. Temporal Fusion Architectures

Temporal fusion architectures in autonomous driving aim to integrate multi-sensor data across different time steps to construct dynamic environmental models, thereby enhancing the continuity of perception and decision-making. The core idea is to utilize historical information to augment the interpretation of the current frame, mitigating issues such as sensor noise, object occlusion, and motion blur, while capturing object dynamics through spatiotemporal modeling. This technology has evolved from traditional signal processing to deep learning-based methods, and more recently, to unified multimodal spatial representations, forming increasingly efficient and robust fusion paradigms.

Early temporal fusion approaches relied on classical algorithms such as Kalman filters and Bayesian filters, which predict object trajectories based on motion models and iteratively refine estimates using sensor observations. For example, the FaF [121] algorithm aligns multi-frame LiDAR point clouds into a common coordinate system, applies voxelization to compress height dimensions into feature channels, and fuses temporal features via 2D convolution along the time axis. These methods avoid the computational cost of 3D convolutions but are constrained by fixed motion model assumptions, making them inadequate for modeling nonlinear motion in complex scenes. Another strategy is late fusion, which first performs object detection on individual frames, then applies data association algorithms (e.g., Hungarian matching) for cross-frame tracking. While computationally efficient, such frame-by-frame processing can lead to error accumulation: Missed or false detections directly impact downstream tracking and prediction accuracy. To address this, recent research has sought to integrate detection, tracking, and prediction into end-to-end networks, reducing inter-module error propagation through joint optimization.

With the advent of recurrent neural networks (RNNs) and long short-term memory (LSTM) networks, temporal fusion shifted toward data-driven feature learning. RNNs propagate historical information through hidden states, while LSTMs introduce gating mechanisms (input, forget, output gates) to dynamically regulate information flow, mitigating vanishing gradient issues. For instance, in lane-change scenarios, LSTMs can infer steering and acceleration strategies from lane markings and adjacent vehicle speeds in previous frames. Tesla’s full self-driving (FSD) system further proposes a spatial RNN module that maps vehicle motion into a 2D grid of hidden states, where each grid cell corresponds to a specific spatial region, with separate channels encoding features such as road edges, lane lines, and obstacles. Only the grid regions relevant to the current field of view are updated, while occluded areas are frozen until visibility resumes, significantly enhancing robustness to dynamic occlusions. Tesla also introduces a temporal feature queue (e.g., buffering features at 27 ms intervals) for handling brief occlusions, and a spatial feature queue aligned with driving distance to retain information in static scenes (e.g., while waiting at a red light).

The bird’s eye view (BEV) space offers a unified framework for multimodal temporal fusion. BEVDet4D [108] retains intermediate BEV features from historical frames, aligns them with the current frame, and applies convolutional layers to extract temporal cues, simplifying velocity estimation into position offset regression. PETRv2 [122] uses 3D positional encoding and pose transformation to align coordinate systems across frames, enabling cross-frame feature interaction via transformer-based cross-attention. These approaches embed temporal information into BEV space, forming 4D (spatial + temporal) feature tensors that support tasks such as velocity estimation and occlusion-aware prediction. For instance, FIERY [123] integrates monocular data and motion priors in BEV space, using probabilistic heatmaps to infer the potential presence of occluded obstacles. To improve efficiency, methods like QTNet [124] introduce sparse query mechanisms: query vectors are generated only for foreground objects (e.g., vehicles, pedestrians), and explicit motion compensation aligns queries across frames, reducing redundant computation on background regions.

Spatiotemporal alignment of multimodal data is foundational to temporal fusion. Raw sensor outputs from LiDAR, millimeter-wave radar, and cameras must be time-synchronized via hardware-software cooperation and transformed into a unified coordinate frame with ego-motion compensation. For example, instantaneous velocity estimates from radar can be fused with historical camera-based object trajectories to construct multi-scale temporal features, enhancing lane-change intent recognition. From a safety redundancy perspective, heterogeneous sensor outputs must undergo cross-validation: if LiDAR and vision localization results deviate beyond a threshold, the system switches to a fallback model and activates redundant control units. Tesla’s occupancy network innovatively replaces traditional 3D detection boxes with voxel-wise occupancy prediction, dividing 3D space into uniform grids and directly determining whether each voxel is occupied. This allows the system to detect irregular obstacles (e.g., overturned vehicles, scattered debris). Inspired by robotic occupancy grid mapping, the network dynamically updates voxel occupancy probabilities via spatiotemporal convolution over video streams.

Despite these advances, temporal fusion still faces challenges in balancing computational cost with real-time performance. Techniques such as sliding window segmentation and dynamic precision adjustment have been proposed to optimize computational load—adapting the number of fused frames or model resolution based on road complexity to achieve millisecond-level response on embedded platforms. Long-term dependency modeling also requires the integration of kinematic priors: For instance, Kalman filtering can suppress sensor noise, while vehicle dynamics models can constrain trajectory predictions to physically plausible paths. In the future, as end-to-end architectures mature, perception, prediction, and planning modules may be jointly optimized within a unified temporal framework, further minimizing information latency and error accumulation.

### 3.3. Applications of 3D Object Detection

Three-dimensional object detection technology has been widely applied in various fields, and in the context of rapidly advancing automation and intelligent technologies, it has become a core tool in many research studies. As the technology continues to evolve, 3D object detection demonstrates enormous potential in object recognition, spatial analysis, and environmental monitoring. This technology not only provides precise spatial data but also achieves efficient target recognition and measurement in complex application scenarios. Whether through laser scanning, unmanned aerial vehicle (UAV) technology, multi-sensor fusion, or the integration of deep learning methods, these techniques are capable of delivering real-time, high-precision detection in dynamic and complex environments.

With the development of smart and precision agriculture, 3D object detection technology has gradually emerged as an essential tool in agricultural research, particularly in crop and vegetation phenotyping and production processes. It not only supplies accurate data for quantitative analyses of crop growth but also enables automation and efficiency in breeding, pest and disease monitoring, and nutrient management. Techniques involving laser scanning, UAVs, multi-sensor fusion, and deep learning have proven effective in real-time target detection and precise measurement in complex agricultural environments. As research deepens, 3D object detection technology has gradually transitioned from laboratory experiments to practical applications, playing a crucial role in monitoring the growth of various crops. For instance, the 3D reconstruction technology for wheat plants, which utilizes point cloud data and virtual design optimization, enables the automated extraction of phenotypic parameters such as plant height and crown width (R^2^ = 0.90, NRMSE = 8%), significantly improving breeding efficiency [125]. The nursery tree point cloud classification model D-PointNet++ achieved 89.90% mIoU in trunk-crown segmentation, providing a novel tool for assessing seedling quality [126]. A mobile surface estimation method for rice height distribution has overcome the challenges posed by interlaced branches and leaves, reaching a height prediction accuracy of 90% and offering prospective data for the feed control of combine harvesters [127]. Future work should focus on developing cross-scale modeling algorithms to enable multi-level structural analysis from individual plants to entire fields. The integration of laser scanning sensors with kernel density estimation techniques [128,129] has facilitated the spatial analysis of strawberry distribution density (grouping accuracy > 94%), providing critical parameters for the development of variable sprayers. Studies have confirmed that laser point cloud data can precisely capture complex canopy structures (similarity > 0.85), and kernel density estimation with a 200-pixel bandwidth can effectively guide the dynamic adjustment of pesticide dosage. In the future, the integration of UAV and ground-based operations could facilitate the construction of 3D thermal maps of canopy pest and disease distributions, enabling targeted spraying and early warning of infestations. Furthermore, nitrogen content monitoring technology based on UAV multi-sensor image fusion [130] has been successfully applied in rice cultivation. By combining RGB and multispectral data, it achieves high-precision estimation of leaf nitrogen content (R^2^ = 0.68, RMSE = 11.45%). This approach significantly enhances model robustness through the fusion of multiple features (spectral bands, vegetation indices, texture), providing real-time data support for precision fertilization. Future extensions of this technology could include major crops such as wheat and maize, integrating Internet of Things (IoT) technologies to construct dynamic nutritional maps and drive the intelligent upgrade of variable fertilization systems.

In addition, deep learning models have achieved significant progress in target detection under complex conditions. An improved YOLOv8 model has demonstrated success in maturity classification for lotus seedpod instance segmentation (mAPmask reaching 98%), and its 3D point cloud reconstruction provides pose estimation for picking robots [131]. A lightweight TS-YOLO model supports real-time detection of tea tips under all illumination conditions (82.12% AP, 11.68 FPS), breaking through the technical bottleneck of nighttime harvesting [132]. To address occlusion issues, a multi-class detection model for cherry tomatoes [133] utilizes a feature enhancement network to distinguish between different types of occlusion by branches and leaves, achieving accuracies of 95.86% (non-occluded) and 84.99% (leaf-occluded), thereby laying the groundwork for optimizing robotic arm obstacle avoidance strategies. Future efforts should further integrate lightweight models [134] and fuse multimodal sensors (e.g., tactile and thermal imaging) to enhance the separation and recognition of densely clustered crops. Inspired by multi-sensor fusion-based 3D object detection, other researchers have also achieved promising results. For example, the integration of RGB-D cameras with attention mechanisms has enabled the detection of peach blossom density [135]; the fusion of computer vision and electronic nose data (using SVR models [136,137]) has improved the accuracy of tomato maturity classification; and the combination of surface-enhanced Raman spectroscopy (SERS) with CNN-LSTM has enabled the detection of trace toxins in corn oil [15].

## 4. Future Directions

In recent years, deep learning-based 3D object detection has achieved remarkable breakthroughs driven by advancements in sensor modality fusion and algorithmic architecture innovation. RGB-based methods have narrowed the accuracy gap with LiDAR approaches in complex scenarios through optimized monocular depth estimation, enhanced stereo matching, and pseudo-point cloud generation. Point cloud processing techniques have balanced efficiency and precision via sparse convolution, hybrid point-voxel modeling, and transformer architectures. Multimodal fusion technologies have established robust cross-sensor perception frameworks through data-level spatial alignment, feature-level cross-modal interaction, and decision-level complementary verification. From an architectural perspective, traditional convolutional networks laid the foundation for real-time detection through voxelization and pillar-based feature compression. BEV methods achieved global perception across multi-camera systems via perspective transformation and spatiotemporal attention mechanisms, while occupancy techniques broke the limitations of bounding boxes through dense voxel modeling. Temporal fusion architectures significantly enhanced understanding of dynamic traffic scenarios through cross-frame motion compensation and feature association. These technological advancements have been successfully applied to practical applications such as crop phenotyping in agricultural robotics and obstacle detection in autonomous driving, accelerating the transition of intelligent systems from laboratory validation to industrial deployment.

Future research should focus on breakthroughs in depth perception enhancement, efficient point cloud processing, and open-scene modeling. For monocular vision’s depth ambiguity, physically constrained multi-scale depth estimation models combined with neural radiance fields (NeRF) could achieve geometrically consistent feature projection, while dynamic weight allocation mechanisms may address sensor failure scenarios. Point cloud processing requires designing highly parallelized sparse convolution operators and graph neural network architectures, optimized through quantization-aware training and dynamic voxel resolution adjustment for embedded platforms. Open-scene understanding demands hybrid neuro-symbolic reasoning systems that integrate kinematic constraints into spatiotemporal transformers to improve physical plausibility in long-term trajectory prediction, alongside hierarchical voxel compression algorithms and event-driven update strategies for large-scale dynamic environments. Cross-domain generalization capabilities can be enhanced through domain-invariant feature disentanglement and synthetic data generation via virtual engines, coupled with multi-task self-supervised learning frameworks to reduce annotation dependency. System-level optimization should explore sensor-algorithm co-design, such as hardware integration of photonic chip LiDAR with programmable ISP cameras, while developing lightweight models via knowledge distillation to meet stringent automotive-grade requirements for low latency and high reliability. With continuous breakthroughs in core technologies, 3D object detection will achieve millimeter-level precision in complex dynamic environments, providing robust environmental cognition for emerging fields like smart agriculture and industrial inspection.

## Figures and Tables

**Figure 1 sensors-25-03668-f001:**
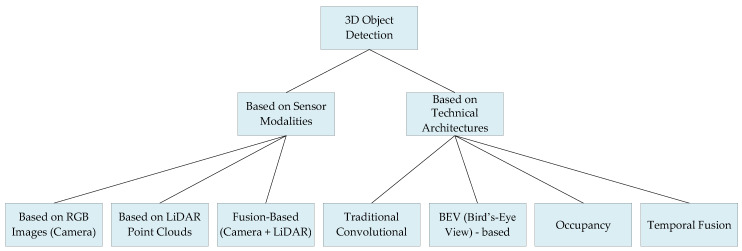
Schematic diagram of sensor modality-technical architecture; for details, refer to Section 3 of this review.

**Figure 2 sensors-25-03668-f002:**
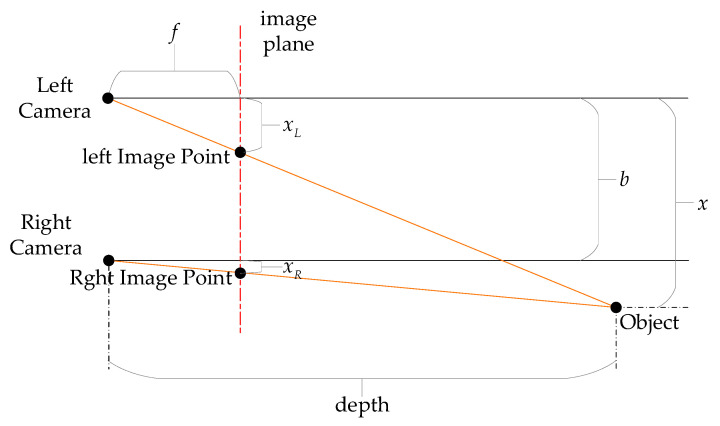
The principle of disparity in a stereo camera system: A single object is simultaneously observed by the left and right cameras, forming image points on their respective image planes with horizontal coordinates $x_{L}$ and $x_{R}$. The baseline distance between the two cameras is denoted as *b*, and the focal length of the cameras is *f*. By computing the disparity $d=x_{L}-x_{R}$ between the corresponding image points, the depth (i.e., the distance from the object to the camera) can be estimated using triangulation formula.

**Figure 3 sensors-25-03668-f003:**
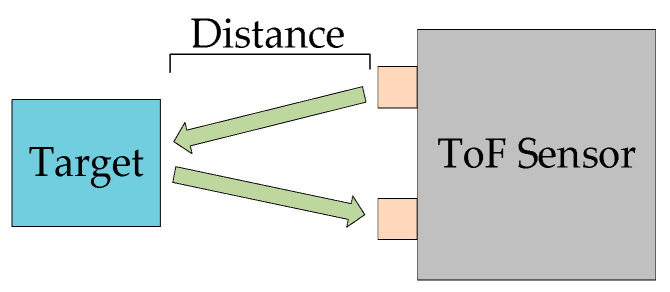
The basic principle of Time-of-Flight (ToF) distance measurement used in LiDAR systems. A ToF sensor emits a light signal (such as a laser pulse) toward a target object. The signal reflects off the object and returns to the sensor. By precisely measuring the time *t* it takes for the light to travel to the object and back, and knowing the speed of light *c*, the distance *D* between the sensor and the object can be calculated using formula.

**Figure 4 sensors-25-03668-f004:**
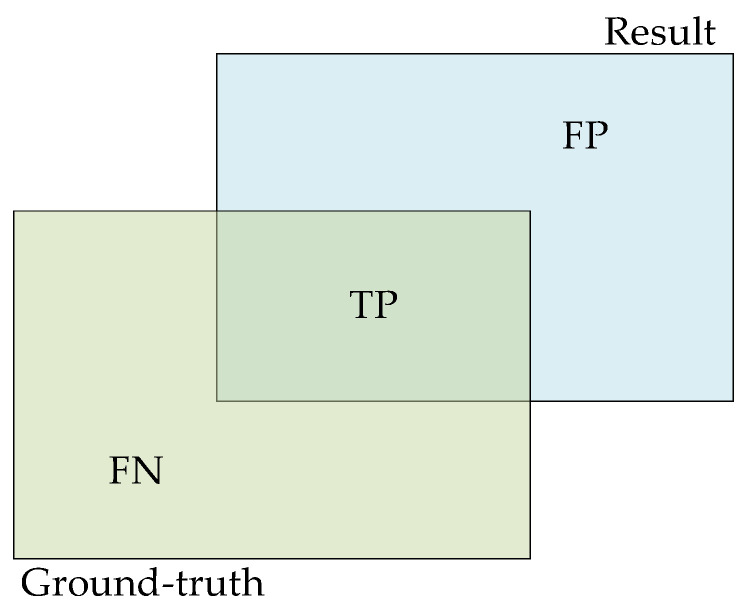
Illustration of *intersection over union* (*IoU*).

**Figure 5 sensors-25-03668-f005:**
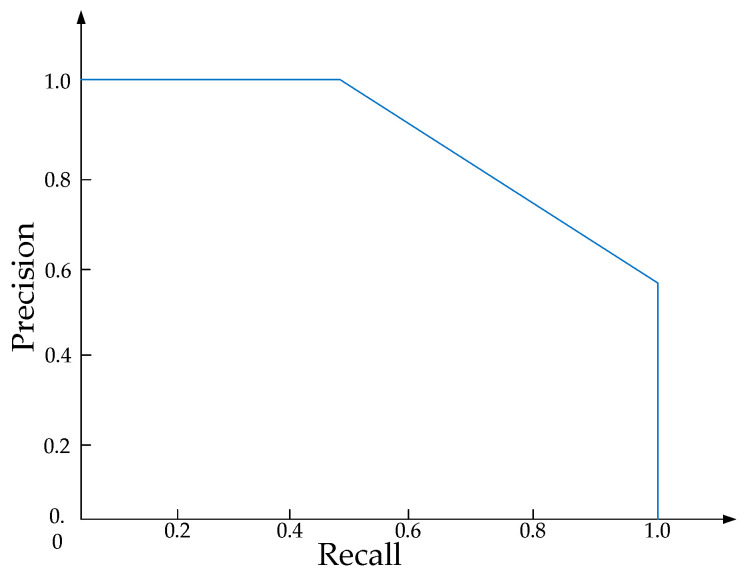
Precision–recall (P-R) curve. The area under the curve, enclosed by the precision-recall curve and the axes, represents the *average precision* (*AP*).

**Figure 6 sensors-25-03668-f006:**
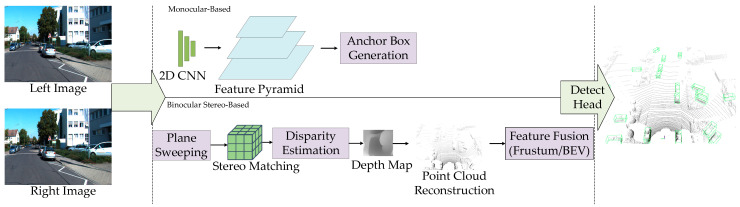
3D object detection based on RGB images. The upper part illustrates the basic pipeline of monocular methods, which involves extracting a feature pyramid through a 2D convolutional neural network and generating candidate anchor boxes. The lower part depicts the processing flow of stereo-based methods, where depth maps are obtained via planar sweeping and disparity estimation, followed by point cloud reconstruction and fusion with image features. Ultimately, both approaches produce 3D detection results through a unified detection head.

**Figure 7 sensors-25-03668-f007:**
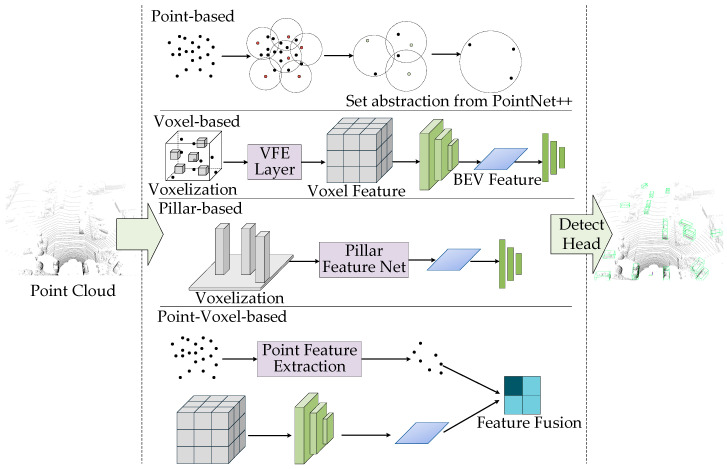
3D object detection methods based on LiDAR point clouds. This figure illustrates the feature extraction and modeling pipelines of four mainstream 3D object detection approaches based on raw point clouds: (1) Point-based methods (e.g., PointNet++) directly sample local regions from the raw point cloud and extract point-wise features; (2) voxel-based methods convert point clouds into voxel representations, extract voxel features, and construct bird’s-eye view (BEV) representations for detection; (3) pillar-based methods extract features using pillar-shaped voxels, offering a balance between accuracy and computational efficiency; (4) point-voxel-based methods jointly extract and fuse both point-level and voxel-level features to enhance overall perception capability. All methods ultimately output 3D bounding boxes through a unified detection head.

**Figure 8 sensors-25-03668-f008:**
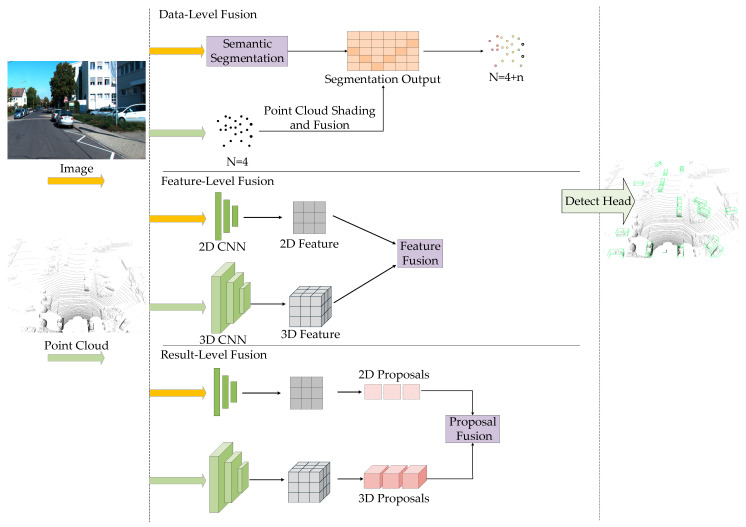
Fusion-based 3D object detection methods. This figure illustrates three levels of fusion strategies between images and point clouds: (1) Data-level fusion enhances the semantic density of point clouds by incorporating image-based semantic segmentation, achieving early fusion at the input stage; (2) feature-level fusion extracts features separately from images and point clouds using 2D and 3D convolutional neural networks, respectively, and then fuses them in the intermediate layers through concatenation or weighted operations; (3) result-level fusion independently generates candidate bounding boxes from each modality and merges them during the post-processing stage. These three strategies differ in terms of the depth of information integration, computational complexity, and final detection accuracy.

**Table 1 sensors-25-03668-t001:** Comparison of Four Sensors.

Device Type	Advantages	Disadvantages
Monocular Camera	Structurally simple and cost-effective, suitable for robotics and UAV applications, but requires algorithmic depth estimation [21].	Lacks direct depth measurement capability, sensitive to lighting/texture variations, and degrades in low-light conditions.
Stereo Camera	Calculates depth directly via binocular disparity, offers high resolution, and adapts to indoor/outdoor environments (active variants enhance performance with infrared illumination).	High computational complexity, texture-dependent matching, baseline-limited range, and reduced efficacy in low-texture or high-glare scenarios.
Mechanical LiDAR	Provides millimeter-level accuracy and 360° scanning, with mature technology and stable point clouds for early autonomous systems.	Bulky size, high cost/power consumption, short lifespan due to mechanical wear, and challenges in mass production.
Solid-State LiDAR	Compact and vibration-resistant, purely solid-state designs eliminate moving parts, ideal for vehicular and robotic navigation.	Pure solid-state variants have limited detection range (e.g., 20–50 m), while hybrid versions retain micro-mechanical components, compromising long-range accuracy.

**Table 3 sensors-25-03668-t003:** Comparison of Sensor Modality-Driven 3D Detection Methods.

Sensor Type	Subcategory	Data Format	Advantages	Limitations
RGB Images	Monocular Vision	Two-dimensional pixel matrix	Low cost, rich texture semantics	Depth ambiguity, scale uncertainty
	Stereo Vision	Stereo image pairs	True depth perception	High computation, texture reliance
LiDAR	Point-based	Raw point clouds (N × 3)	Preserves geometric details	High computational complexity
	Voxel-based	Three-dimensional voxel grids	Structured representation	Quantization errors
	Pillar-based	Two-dimensional pseudo-images	Balances efficiency and structure	Compressed height information
Multimodal	Data-Level Fusion	Raw sensor alignment	Preserves raw information	Requires precise calibration
	Feature-Level Fusion	Cross-modal features	Complementary strengths	Feature alignment challenges
	Result-Level Fusion	Independent outputs	Flexibility	Redundant computation

**Table 4 sensors-25-03668-t004:** Evolution of Technical Architecture-Driven 3D Detection Methods.

Architecture Type	Key Technical Features	Core Innovations	Strengths	Key Challenges
Traditional Conv.	Voxelization + 3D conv.	Structured data representation	Hardware-friendly, real-time	Quantization errors
	Sparse convolution	Non-empty voxel processing	A total of 50%+ efficiency gain	Long-range feature decay
BEV Methods	Perspective transformation	LSS 3D feature lifting	Unified multi-view perception	Projection distortion
	Spatiotemporal attention	Cross-frame alignment	Dynamic tracking	Historical frame storage
Occupancy	Dense voxel modeling	Three-dimensional spatial probability prediction	Irregular object detection	O(n^3^) complexity
	NeRF fusion	Joint NeRF + occupancy training	High-fidelity reconstruction	Data-hungry
Temporal Fusion	RNN/LSTM	Sequential modeling	Motion trajectory prediction	Long-term dependency issues
	Four-dimensional feature tensor	BEV + temporal dimension	Velocity estimation (<0.3 m/s)	High memory usage

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
