# Peer review of "A Survey of Deep Learning-Driven 3D Object Detection: Sensor Modalities, Technical Architectures, and Applications"

_sensors, 2025, doi:10.3390/s25123668_

Round 1
Reviewer 1 Report
Comments and Suggestions for Authors
The paper reviews the field of 3D object detection. Although I appreciate the work the authors have already done, I am quite critical about this survey, because: (1) it is not compared with other existing surveys, (2) the methods are categorized using just a single taxonomy, and (3) it is not clear how the included papers were selected. Please find the detailed comments below.
1) The introduction lacks a brief overview of the paper content and structure.
2) In Section 3, the methods are categorized by the sensor type used to acquire the input data. However, it is not clear how the papers included in the survey were selected - there are a few methodologies that can be employed for this purpose (like systematic literature review), but the authors don't mention that - it looks as if they just included a random bunch of papers that were published on 3D object detection.
3) YOLO V4-Tiny (ref [3]) is not the best example for a CNN - if the authors want to provide examples, these should be better selected to showcase some ground-breaking works. This also concerns other examples provided throughout the paper.
4) The field of 3D object detection is intensively explored and there are also many recent surveys on this topic. The authors mention two of them (by Arnold [1] and Mao [34]), but they are even not presented as surveys ([1] serves as an example and [34] to provide a reference for an equation). So what's the difference between your survey and the existing surveys? Does really the community need yet another survey on the same topic? Why should someone read your survey, rather than one of other recent surveys? These are basic questions and every survey should begin with answering them. The differences between the existing surveys and yours should be first concisely outlined in the introduction and then a separate section should be devoted to explaining these differences (among the existing surveys, as well as between them and yours).
5) Section 2 showcases the existing sensors that provide input data for 3D object detection. There are a few randomly selected photos of LiDARs and a camera - that's redundant, everyone knows how these devices look like. Instead, there should be some meaningful diagrams and figures to better explain the specifics of these sensors.
6) The evaluation metrics are explained in detail, but I would suggest to add some visual examples to better explain the mAP metric, as (according to my observations) it is not intuitive and easy to understand. Providing such extended explanations may be very helpful.
Reviewer 2 Report
Comments and Suggestions for Authors
In this paper, the authors focused on 3D object detection, and presented a review of the existing methods for such tasks (for different data modalities). Indeed, the topic is worthy of investigation, and it easily falls into the scope of the journal. However, the manuscript suffers from the following shortcomings that should be thoroughly addressed before it could be, in my opinion, considered for publication:
- The authors should make sure that each abbreviation is defined at its first use. I spotted some undefined abbreviations around the manuscript (even in the abstract).
- Although the English is acceptable, the manuscript would greatly benefit from careful proofreading – there are some inconsistencies around the manuscript (e.g., concerning upper/lowercasing – again, I spotted such inconsistencies in the abstract). Also, please carefully revise whitespaces around the manuscript (there are some missing whitespaces, especially near the references).
- It might be useful to expand the state of the art review to include other features of the existing methods, e.g., their robustness against low-quality data or missing modalities (in the case of multi-modal processing). Discussing such practical issues could be of huge importance to the community.
- It would be useful to announce the structure of the manuscript in the introductory section.
- Table 1 is rather difficult to grasp, it would be good to add some horizontal lines to divide it into specific subsections which would correspond to the discussed methods.
- In the table summarizing the dataset, it would be great to also add a year of publication (overall, the year of publication should be added in all tables).
- While discussing the datasets (in the table), it would be very useful to indicate specific links to these datasets, information whether they contain training/test dataset splits, and what are the current state of the art methods for these particular problems.
- The quality of the figures should be improved – all of them should be high-resolution in a vector format.
- Some sections are extremely lengthy and thus difficult to read and follow. I encourage the authors to split them into subsections to enhance their readability, and to make understanding them (and searching for particular bits of information) easier and more straightforward.
Reviewer 3 Report
Comments and Suggestions for Authors
(1) Sometimes "LiDAR" is referred to as "LiDAR" and other times as "lidar" in the text, especially within citations. To maintain professional consistency throughout the paper, it is suggested to unify the terminology, preferably using "LiDAR" consistently across all sections, figures, and references.
(2) Some figure captions, such as those for Figures 1 to 5, are too brief, for example, simply stating "Monocular Camera" or "Fusion-Based 3D Object Detection Methods" without context. It would be better to add short but meaningful descriptions that explain what the figure illustrates and how it connects to the main discussion.
(3) Minor grammatical polishing would improve the overall fluency of the text; for instance, sentences like "By combining the coverage from the two beams, the system enables retrieval of the horizontal wind field" could be rephrased more smoothly as "This dual-beam coverage enables horizontal wind field retrieval," and similar style refinements should be considered throughout the manuscript.
(4) In the dataset comparison section, the KITTI, nuScenes, and Waymo Open Dataset descriptions are quite detailed, while the discussion on ONCE and Argoverse 2 is relatively brief; it would improve the balance and completeness of the review if these two datasets could be described with slightly more detail and analysis.
(5) Some citations are incorrectly formatted without a space before the bracket, such as "smart agriculture[9–11]"; this should be corrected to "smart agriculture [9–11]" consistently throughout the manuscript to meet formatting standards and improve readability.
(6) The connection between Section 2 (Basic Knowledge) and Section 3 (Detection Methods) feels slightly abrupt; adding a short bridging paragraph to summarise how the understanding of sensor characteristics and dataset structures leads naturally into detection methodology discussions would greatly enhance the logical flow.
(7) Challenges such as depth estimation difficulties for monocular images and sparsity issues for LiDAR data are mentioned multiple times across sections; it would be helpful to summarise the main challenges earlier, perhaps towards the end of the Introduction or the beginning of Section 2, to give readers a clearer roadmap of the problems being addressed.
(8) Although equations like (3) and (4) are well explained, in later parts of the paper such as around equation (9), it would be helpful to provide brief reminders or explanations inline for key symbols (such as mTP and mAP), so that readers can better follow the mathematical derivations without needing to refer back to earlier definitions.
(9) Given the dense technical content, the paper would benefit from inserting one or two additional visual summaries, such as a table comparing the advantages and disadvantages of monocular, stereo, and LiDAR-based 3D object detection methods, or a diagram summarising sensor fusion strategies, to help readers grasp key points more efficiently.
(10) Minor typographical and formatting issues exist throughout the manuscript, such as inconsistent use of capitalisation ("Fusion-based methods" vs "fusion-based methods") in the abstract and body text; a careful proofreading pass is recommended to eliminate these small errors and ensure professional polish.
Comments on the Quality of English LanguageMinor grammatical polishing would improve the overall fluency of the text; for instance, sentences like "By combining the coverage from the two beams, the system enables retrieval of the horizontal wind field" could be rephrased more smoothly as "This dual-beam coverage enables horizontal wind field retrieval," and similar style refinements should be considered throughout the manuscript.
Round 2
Reviewer 1 Report
Comments and Suggestions for Authors
The authors have addressed some of my comments, but my main concern is that I cannot see any reason why someone would read your survey rather than another one. The previous surveys are enlisted in Section 1, but they are criticized in unacceptable way. For example, the recent survey by Wang et al. is claimed to have a (too) complex structure. A good survey presents the state of the art from different perspectives and that's exactly what that survey does and yours fails to do.
The categories are presented in Table 3, but they mix different dimensions (sensor type and fusion type - these categories are orthogonal). This should be thoroughly reworked and the methods should be presented considering different categorization perspectives.
The authors claim to have added a subsection entitled "Literature Selection Methodology" on page 8. In fact, there is no such subsection, but I found the added information on page 15. Nevertheless, the methodology is quite vague, Google Scholar should not be the primary database; how about other publishers like Elsevier (Science Direct)? Also, the fixed time frame is not reasonable, as there may be many earlier works that are still relevant and should not be missed.
The mAP is not well explained - it does not mention the (very important) issue of setting the IoU threshold.
Reviewer 2 Report
Comments and Suggestions for Authors
Thank you for addressing my concerns.
Author Response
Thank you for your time and input. Wishing you and your loved ones health and prosperity.
Reviewer 3 Report
Comments and Suggestions for Authors
Since the author has addressed the comments and improved the submission, publication is recommended.
Author Response

(The authors gave the same response as above.)
